# ENHANCING GRAPH GENERATION WITH FIRST-ORDER LOGIC RULES

## ABSTRACT

Existing graph generative models produce graphs that are often quite realistic, but sometimes miss domain-specific patterns. Enhancing graph learning with domain knowledge is one of the current frontiers for neural models of graph data. In this paper, we propose a new approach to enhancing deep graph generative models with knowledge that is represented by first-order logic rules. First-order logic provides an expressive formalism for representing interpretable knowledge about relational structures. Our conceptual contribution is a new first-order semantic loss function for training a graph generative model on relational data: maximize the model likelihood subject to a *moment matching constraint*, namely that the expected instance count of each rule matches its observed instance count. Our algorithmic contribution is a novel method for computing the expected instance count of a first-order rule for a standard generative mixture model based on matrix multiplication. Empirical evaluation on seven benchmark datasets, both homogeneous and heterogeneous, shows that moment matching improves the quality of generated graphs substantially (by orders of magnitude on standard graph quality metrics), and improves predictive accuracy on the downstream task of node classification.

## 1 INTRODUCTION

Generative models for graphs based on graph neural networks (GNNs) have achieved great success in modeling complex graphs (Hamilton, 2020). One of the current research frontiers is enhancing graph learning with domain knowledge (Tian *et al.*, 2024) (Wang *et al.*, 2020) (Sun *et al.*, 2021) (Niresi *et al.*, 2024) (Yu *et al.*, 2023) (Agarwal *et al.*, 2022). Different enhancement methodologies are appropriate for different types of knowledge. In this paper, we consider leveraging knowledge in the form of a *first-order logic knowledge base* (Russell and Norvig, 2010), comprising a set of first-order (FO) formulas. FO formulas represent domain knowledge by specifying important patterns in a domain. Because formulas used in knowledge representation practice often take the form of if-then rules, we refer to our approach as rule-enhanced graph generation. An example rule would be "If person $X$ works in city $Y$, then $X$ lives in city $Y$ (with probability $p$)".

**Advantages.** Logical formulas have several advantages for enhancing graph learning. (1) *Expressiveness*: First-order formulas are one of the most common formalisms for representing domain knowledge in AI and database systems (Russell and Norvig, 2010). (2) *Interpretability*: Logical formulas are easily understood by users and domain experts. (3) *Learnability*: The field of statistical-relational learning (SRL) has developed methods for learning relevant formulas from a heterogeneous training graph, known as *structure learning*. (4) *User Control*: Users can control the behavior of the final graph generation system in a mixed-initiative approach, by specifying and/or rejecting formulas. (5) *Graph Realism and Data Efficiency*: Matching first-order formulas leads to generating more realistic graphs, while requiring less training data.

**Approach.** Figure 1 shows our system components. We show how fundamental ideas from SRL (Raedt *et al.*, 2016) can be combined with deep graph generative models (GGMs). A fundamental concept of SRL is *moment matching* (Domingos and Lowd, 2019; Russell, 2015; Kuzelka *et al.*, 2018). The general idea is that a formula can be viewed as specifying a *motif* or subgraph pattern with an **instance count** in a given graph. Formula moment matching requires that for each formula,

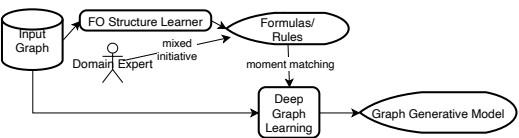

Figure 1: System Overview for Rule-Enhanced Graph Generation

*the expected instance count for a model should match the observed instance count in a training graph.* Our novel GGM training objective is to *maximize the GGM likelihood subject to moment matching*.

Our algorithmic contribution is a *differentiable new matrix multiplication method* for computing observed and expected instance counts. We show that for every conjunctive formula (satisfying a minor syntactic constraint), there is a corresponding sequence of adjacency matrices, such that i) the observed instance count is obtained by multiplying the data adjacency matrices, and ii) the expected instance count for a standard mixture model with conditionally independent links is obtained by multiplying expected adjacency matrices.

**Evaluation.** Our methodology uses an A-B design where we compare training a recent state-of-the-art variational graph auto-encoder (Mahmoudzadeh *et al.*, 2024), called VGAE+, with and without moment matching, on seven benchmark datasets. We find that rule-enhanced VGAEs score better than standard VGAEs on several metrics: (1) They generate *more realistic graphs*, by orders of magnitude, as measured by SOTA graph quality metrics (F1 MMD) (Thompson *et al.*, 2022; O'Bray *et al.*, 2022). (2) On the downstream task of *node classification*, the rule-enhanced VGAE node embeddings improve accuracy compared to standard VGAE. (3) Learning curves for node classification show that first-order domain knowledge often leads to more data efficient learning.

**Contributions** Our main contributions can be summarized as follows.

- A new semantic loss objective function for enhancing generative graph training with domain knowledge represented by first-order formulas: Maximize the data likelihood of a graph generative model, constrained so that the observed number of formula instances matches the expected number of formula instances.

- A new matrix multiplication algorithm for counting the number of formula instances in a graph.

- A proof that the matrix multiplication algorithm can also be used to estimate the expected number of instances for a standard mixture model. It can therefore be leveraged to compute the new semantic loss objective.

- Our new VGAE+R system uses the new objective function to train a VGAE+ model that matches formula instance counts.

## 2 RELATED WORK

Our work falls under the heading of *neuro-symbolic AI*, a cutting-edge field of AI that aims to combine symbolic formalisms, such as first-order logic, with neural network learning; see Figure 2. For surveys of neuro-symbolic AI, please see (Raedt *et al.*, 2020; Garcez and Lamb, 2023), and Kautz's 2022 Engelmore lecture. Within Kautz's taxonomy, our approach belongs to the *semantic loss* frameworks (type 5) where symbolic knowledge is encoded into the network's loss function (Kautz, 2022; Xu *et al.*, 2018; Marra *et al.*, 2019). The trained system is a standard NN model that does not utilize rules at test time. In contrast, reasoning approaches typically perform symbolic inference (Raedt *et al.*, 2020; Qu *et al.*, 2021) at test time.

Compared to previous semantic loss approaches (Xu *et al.*, 2018), our main innovation is that *we incorporate knowledge expressed in first-order logic, rather than the less powerful formalism of propositional logic*. For example, a propositional rule would be "if a movie is a horror movie, it is not likely to be a romance". A first-order rule could be "if a user rates a horror movie, the user is most likely to be a man". Since first-order rules incorporate relationships, *first-order logic (FOL) can*

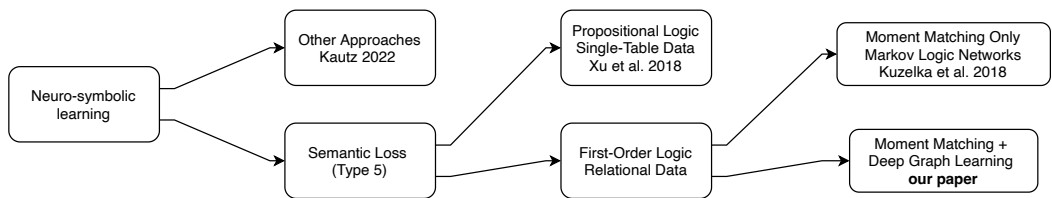

Figure 2: Within neuro-symbolic AI, we develop a new first-order semantic loss approach.

*leverage the full power of relational data.* We show that our semantic loss for FOL rules reduces to the loss of (Xu *et al.*, 2018) in the propositional case. Compared to previous FOL approaches (e.g., Marra *et al.* (2019)), we use standard FOL semantics (not fuzzy logic), and our computations do not require as input the full Cartesian product grounding over all domain elements (nodes).

*Markov Logic Networks and Maximum Entropy Moment Matching.* We use the same FO knowledge representation structure as the well-known **Markov Logic Network** (MLN) model, namely a set of FO formulas. The MLN formalism has been applied to represent knowledge in a number of domains, and it has sufficient expressive power to capture other FO formalisms, such as rule-based knowledge (Domingos and Lowd, 2019).

In terms of model training, Kuzelka *et al.* (2018) show that a distribution $P$ over graphs maximizes entropy subject to moment matching if and only if $P$ is defined by an MLN with maximum likelihood weights. Both the maximum entropy objective and our constrained likelihood objective capture the global graph statistics represented by instance counts. However, the GGM likelihood can in addition capture local graph patterns. For example, matching the number of observed triangles in a graph is unlikely to capture community structure, or which nodes have special properties such as centrality.

*Deep graph generative models.* The closest predecessor to our work is the constrained VGAE model of Ma *et al.* (2018) where a VGAE likelihood is maximized subject to a constraint of the form $g(\boldsymbol{\theta}) = 0$. While this general form covers moment matching, the work of Ma *et al.* does not incorporate FO logic for specifying graph patterns, nor does it address computing pattern counts.

In principle the moment matching likelihood objective can be used for maximum likelihood training with any deep graph generative model. We selected VGAEs as our base model for several reasons. (1) They are a well-established and widely used GGM. Mahmoudzadeh *et al.* (2024) show that their VGAE+ model is a strong multitask model that provides accurate predictions for a wide range of knowledge graph queries, based on inference from a single model. (2) They support learning from a single large graph, rather than from a set of graphs (Faez *et al.*, 2021). Rule learners also utilize the single-graph setting (Qian and Schulte, 2015; Meilicke *et al.*, 2024), so the VGAE input data are compatible with the rule learner input data. (3) As we show in this paper, the conditional link independence assumptions of VGAEs facilitates the computation of expected rule instance counts. Extending rule moment matching to other generative models is a fruitful topic for future research.

## 3 BACKGROUND ON FIRST-ORDER LOGIC

**Attributed Heterogenous Graphs** An attributed graph is a pair $\mathcal{G} = (V, E)$ where $V$ is a set of nodes of size $|V| = n$ and $E \subseteq V \times V$ is a set of edges. Node features are summarized in $n \times f$ matrix $\mathbf{X}$ and node labels in a $n \times L$ matrix $\mathbf{L}$ where the $u$-th row of $\mathbf{L}$ is a one-hot encoding of the label of node $u$. Different edge types are represented by a set of adjacency matrices $\mathbf{A} = \{\mathbf{A}_1, \ldots, \mathbf{A}_T\}$. The notation $\mathbf{A}_r[u, v] = 1$ indicates that there is a link $u \rightarrow_r v$ of type $r$ from node $u$ to node $v$. Figure 3 shows part of the information in an attributed graph using the tabular SQL format.

| User | | | Rating | | | Movie | | | |
|---|---|---|---|---|---|---|---|---|---|
| User Id | Age | Gender | User Id | Movie Id | Rating | Movie Id | Action | Drama | Horror |
| 3 | 0 (0.34) | M (0.55) | 3 | The Dictator | 1 (0.75) | The Dictator | 0 (0.38) | 0 (0.73) | 0 (0.85) |
| 5 | 1 (0.43) | F (0.34) | 5 | Thor | 4 (0.36) | Thor | 1 (0.49) | 0 (0.66) | 0 (0.4) |
| 7 | 2 (0.90) | M (0.84) | 5 | The Dictator | 3 (0.84) | BraveHeart | 1 (0.91) | 1 (0.41) | 1 (0.7) |
| ... | ... | ... | 7 | BraveHeart | 5 (0.98) | ... | ... | ... | ... |
| | | | ... | ... | ... | | | | |

Figure 3: Excerpt from a relational dataset. (a) An attributed graph represented in table format. (b) The probabilities assigned to each data entry specify a probabilistic graph (see below).

| Conjunction $\phi$ | $n_\phi(\mathcal{G})$ |
|---|---|
| $Age(User) = 0$ | 376 |
| $Rating(User, Movie) = 1$ | 4701 |
| $Age(User) = 0, Rating(User, Movie) = 1$ | 2524 |

Table 1: Conjunction Instance counts in the MovieLens database $\mathcal{G}$

**First-Order Logic**  We follow previous work in SRL (Schulte and Gholami, 2017; Kimmig *et al.*, 2014). A **population** is a set of individuals of the same type (e.g., a set of $Users$, a set of $Movies$). Individuals are denoted by constants (e.g., $user_3$ and $thor$). An attributed graph specifies a set of individuals (nodes) for each type. A **node variable** ranges over a population, and is denoted in upper case such as $User, Movie, U, V$. A unary **functor** maps an individual to a value, and corresponds to a node attribute/label. A binary functor maps an ordered pair of individuals to a value, and corresponds to an edge/edge type. Functors are denoted $f, f'$ etc.

A **first-order** term (FOT) is of the form $f(\boldsymbol{U})$ where each population variable $U_i$ is of the appropriate type. FOT examples are $age(User)$ and $rating(User, Movie)$. A FOT can be instantiated with individual constants, much like an index in a plate model (Kimmig *et al.*, 2014). A **grounding** $\boldsymbol{U} = \boldsymbol{u}$ for a list of FOTs simultaneously replaces each population variable in the list by a constant. (We assume that different population variables are replaced by different constants.) A ground term, Python-style, assigns individuals as argument to node variables, then applies the functor to return a value. Examples are $age(User = user_5)$, and $rating(User = user_5, Movie = thor)$.

An FO **literal** is of the form $\ell \equiv f(\boldsymbol{U}) = v$. A **conjunction** is a list of literals $\phi = \ell_1, \ldots, \ell_s$. We write $\phi(\boldsymbol{U})$ for an FO conjunction and $\phi(\boldsymbol{U} = \boldsymbol{u})$ for a ground conjunction. A graph $\mathcal{G}$ **satisfies** a ground literal if the graph assigns value $v$ to the ground term $f(\boldsymbol{U} = \boldsymbol{u})$, and satisfies the conjunction $\phi$ if it satisfies each ground literal in the conjunction. The **instance count** $n_\phi(\mathcal{G})$ in a graph $\mathcal{G}$ returns the number of $\phi$-groundings satisfied by graph $\mathcal{G}$.

A **probabilistic graph** $\tilde{\mathcal{G}}$ assigns a probability $p_{\tilde{\mathcal{G}}}(\ell(\boldsymbol{U} = \boldsymbol{u}))$ to each ground literal. The **probabilistic instance count** of a conjunction (Kuzelka, 2023) is the probability product, summed over all conjunction groundings:

$$n_\phi(\tilde{\mathcal{G}}) = \sum_{\boldsymbol{U}=\boldsymbol{u}} \prod_{i=1}^{s} p_{\tilde{\mathcal{G}}}(\ell_i(\boldsymbol{U} = \boldsymbol{u})) \text{ for } \phi = \ell_1, \ldots, \ell_s \tag{1}$$

**Examples.**  $Age(User) = 1, rating(User, Movie) = 4$ is an FO conjunction. Its grounding $age(User = user_5) = 1, rating(User = user_5, Movie = thor) = 4$ is satisfied by the data of Figure 3(a). In the probabilistic graph Figure 3(b), the probability of this conjunction is $0.43 \times 0.36 = 0.1548$.

Table 1 illustrates FO instance counts using the MovieLens dataset (Qian and Schulte, 2015). Movielens contains 376 users at age level 0. The number of user-movie pairs with a rating of 1 is 4701. The number of such pairs with the user at age level 0 is 2524. *An FO conjunction specifies a graph motif*, and the instance count is the motif count (Ma *et al.*, 2019) (see Figure 9 for illustration).

## 4 Rule-Enhanced Graph Generation

This section considers how to enhance training a parametrized graph generative model (GGM) $P_{\boldsymbol{\theta}}$ on a training graph $D$, with a list of formulas $\phi_1, \ldots, \phi_k$. Our *semantic loss objective* maximizes the data likelihood $P_{\boldsymbol{\theta}}(D)$, subject to the FO moment matching constraint that $E_{\boldsymbol{\theta}}[n_i] = n_i(D)$, where $n_i(D)$ is the **data instance count** of formula $\phi_i$, and $E_{\boldsymbol{\theta}}[n_i] \equiv \sum_{\mathcal{G}} P_{\boldsymbol{\theta}}(\mathcal{G}) n_i(\mathcal{G})$ is the **expected instance count** for the GGM. For a mixture model GGM, we derive the following Lagrangian ELBO.

**Proposition 1.** *Suppose that $P_{\boldsymbol{\theta}}(\mathcal{G}) = \int p(\mathcal{G}|\boldsymbol{z})p(\boldsymbol{z})d\boldsymbol{z}$ is a mixture model. Then*

$$\ln P_{\boldsymbol{\theta}}(D) - \lambda/k \sum_{i=1}^{k} \rho(n_i(D), E_{\boldsymbol{\theta}}[n_i]) \geq \tag{2}$$

$$E_{\boldsymbol{z} \sim q_{\phi}(\boldsymbol{z}|D)}[\ln P_{\boldsymbol{\theta}}(D|\boldsymbol{z}) - KL\big(q_{\phi}(\boldsymbol{z}|D)||p(\boldsymbol{z})\big) \tag{3}$$

$$-\lambda/k \sum_{i=1}^{k} \rho(n_i(D), E_{\boldsymbol{\theta}}[n_i|\boldsymbol{z}])], \tag{4}$$

*where $\rho(count_1, count_2) \geq 0$ is a differentiable count distance metric convex in $E_{\boldsymbol{\theta}}[n_i|\boldsymbol{z}])]$.*

Proposition 1 says that the constrained likelihood Equation (2) can be approximated by our new **moment matching variational ELBO objective** (3). To compare an expected count to an observed count, our experiments use

$$\rho(n_i(D), E_{\boldsymbol{\theta}}[n_i|\boldsymbol{z}]) = |\ln n_i(D) - \ln E_{\boldsymbol{\theta}}[n_i|\boldsymbol{z}]|.$$

Conjunction counts grow exponentially with the number of node variables in the conjunction. Comparing expected counts on a log-scale decreases the impact of the number of node variables and improves numeric stability. With this choice of $\rho$, *the FO semantic loss Equation* (2) *reduces to the semantic loss of (Xu* et al.*, 2018) for a propositional formula $\phi$*; see Appendix A.6 for details.

### 4.1 IMPLEMENTING THE MOMENT MATCHING ELBO

Our novel VGAE+R architecture extends the recent VGAE+ architecture (Mahmoudzadeh *et al.*, 2024) to match rules, including the new **motif loss** (4).

**Encoder-Decoder Architecture.** Figure 10 shows the VGAE+R architecture. The **encoder** model $q_{\phi}(\boldsymbol{z}|D)$ can be any GNN that maps a heterogeneous graph to node embeddings, such as RGCN. The VGAE+R **decoder** independently maps node embeddings to different graph components with three different decoders (Mahmoudzadeh *et al.*, 2024):

$$\ln P_{\boldsymbol{\theta}}(D|\boldsymbol{z}) = [\alpha \ln p_{\eta}(\mathbf{A}|\boldsymbol{z}) + \beta \ln p_{\psi}(\boldsymbol{X}|\boldsymbol{z}) + \gamma \ln p_{\phi}(\boldsymbol{L}|\boldsymbol{z})$$

where $p_{\eta} : \mathbb{R}^d \times \mathbb{R}^d \to [0, 1]$ is a trainable **link decoder**, $p_{\psi}$ is a trainable **feature decoder**, and $p_{\phi}$ is a trainable **label decoder** (see Figure 10). The hyperparameters $\alpha$, $\beta$ and $\gamma$ weight the importance of different reconstruction tasks.

**Computing Expected Instance Counts.** Given a set of node embeddings $\boldsymbol{z}$, the **expected graph** $\tilde{\mathcal{G}}_{\boldsymbol{z}}$ is a probabilistic graph that assigns a probability to each ground literal by applying the decoder to the relevant links/node features/edge types. For examples see Figure 3 and Figure 11.

**Proposition 2.** *The expected instance count given a set of node embeddings can be computed as the instance count in the expected graph: $E_{\boldsymbol{\theta}}[n_i|\boldsymbol{z}] = n_i(\tilde{\mathcal{G}}_{\boldsymbol{z}})$.*

The proof is in the supplement. The upshot is that *FO moment moment matching can be implemented by performing (probabilistic) instance counting in a single graph.*

## 5 MATRIX MULTIPLICATION FOR INSTANCE COUNTING

SOTA MLN structure learners output a set of conjunctive formulas or if-then rules (Qian and Schulte, 2015; Khot *et al.*, 2011; Cui *et al.*, 2022; Potter *et al.*, 2024). We discuss instance counting for conjunctive formulas, which we can be extended to if-then rules by restricting counts to instances that match the antecedent (body); see Appendix A.4 for more details.

This section presents a novel matrix multiplication method for instance counting with conjunctive formulas, that is differentiable and applies to both discrete and probabilistic graphs. To illustrate the basic idea, consider the conjunction $R(U_1, V_1), R(V_1, V_2), R(V_2, U_1)$, whose instance count gives the number of triangles in an undirected graph represented by an adjacency matrix $\mathbf{A}$. It is well-known

that the triangle count is given by $\sum_{u=1}^{n} \mathbf{A}_{u,u}^3$, the trace of the third power of the adjacency matrix. We generalize this approach to a large class of logical formulas. A **chain conjunction** of binary literals is of the form $\phi = \ell_1(U_1, V_1), \ldots, \ell_P(U_P, V_P)$ where $V_i = U_{i+1}$ for every $i$. Algorithm 1 maps each chain conjunction to a sequence of adjacency matrix multiplications, such that the conjunction's instance count can be found by executing the matrix multiplications.

---

**Algorithm 1** Matrix Multiplication for Instance Counting

1: **Input:** Chain conjunction $\phi = \{\ell_1(U_1, V_1), \ldots, \ell_P(U_P, V_P)\}$
2: **Output:** Instance count $n_\phi(\mathcal{G})$ or expected count $n_\phi(\tilde{\mathcal{G}}_{\boldsymbol{z}})$
3: {Initialize adjacency matrices $\mathbf{A}_{\ell_k}$ for binary literals $\ell_k, k = 1, \ldots, P$ }
4: **for** $k = 1$ to $P$ **do**
5:    **if** positive literal $\ell_k = R(U_k, V_k) = 1$ **then**
6:       $\mathbf{A}_{\ell_k} \leftarrow \mathbf{A}_r$
7:    **else if** $\ell_k = R(U_k, V_k) = 0$ **then**
8:       $\mathbf{A}_{\ell_k} \leftarrow \neg \mathbf{A}_r$ where $\neg \mathbf{A}_r$ is the complement of $\mathbf{A}_r$
9:    **end if**
10: **end for**
11: $O_1 \leftarrow \mathbf{A}_{\ell_1}$
12: **for** $k = 1$ to $P - 1$ **do**
13:    $O_{k+1} \leftarrow O_k \cdot \mathbf{A}_{\ell_{k+1}}$
14:    **if** $V_{k+1} = U_1$ **then**
15:       Zero out the non-diagonal entries of $O_{k+1}$
16:    **end if**
17: **end for**
18: **Return:**
19: $n_\phi(\mathcal{G}) = \sum(O_P(\phi))$ for input graph $\mathcal{G}$
20: $n_\phi(\tilde{\mathcal{G}}_{\boldsymbol{z}}) = \sum(O_P(\phi))$ for expected graph $\tilde{\mathcal{G}}_{\boldsymbol{z}}$

---

**Example.** Consider the chain conjunction
$AdvisedBy(Student, Professor)$, $Teaches(Professor, Course)$ $TakesCourse(Course, Student)$.
Figure 4 shows the corresponding sequence of matrix multiplications in a sample graph.

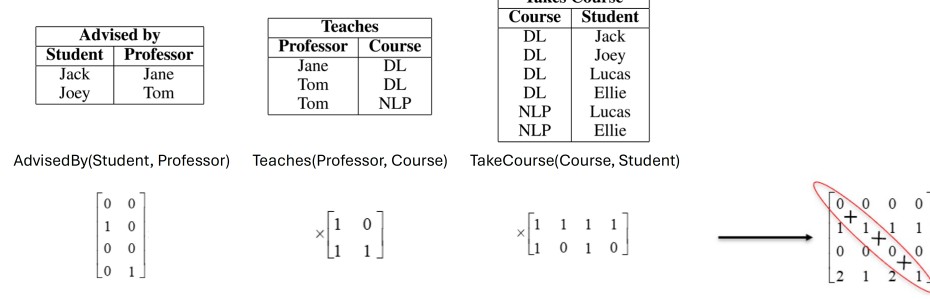

Figure 4: The matrix multiplication sequence for our example conjunction and sample graph data. The final result is 2, which is the number of satisfying groundings in the input graph.

**Extensions.** Unary literals can be included by omitting nodes from the input graph that do not satisfy them. Probabilistic instance counts can be obtained by using soft matrices $\tilde{A}, \tilde{X}, \tilde{L}$; see Appendix A.9.

**Correctness.** The next proposition shows that the instance count for the chain conjunction can be obtained through summing over the entries in the constructed matrix product.

**Proposition 3.** *Let $\phi$ be a centered chain conjunction of length $k$, i.e., the first node variable is the only one that appears twice non-consecutively.*

1. *For an input graph $\mathcal{G}$, the $(u, v)$-th entry of $O_k$ counts the number of groundings of $\phi$ in $\mathcal{G}$ where $U_1 = u$ and $V_P = v$. Therefore $n_\phi(\mathcal{G}) = \sum(O_k(\phi))$.*

2. *For an expected graph $\tilde{\mathcal{G}}_{\boldsymbol{z}}$, the $(u, v)$-th entry of $O_k$ counts the expected number of groundings of $\phi$ where $U_1 = u$ and $V_P = v$. Therefore $n_\phi(\tilde{\mathcal{G}}_{\boldsymbol{z}}) = \sum(O_k(\phi))$.*

In our experiments, we found that all learned rules were centered. The Appendix extends the matrix multiplication method to non-centered chains.

**Computational Complexity**  Algorithm 1 translates a logical formula into a sequence of matrix multiplications in time linear in the length of the formula. The number of binary literals is small enough to be treated as a constant $k \leq 5$. The bottleneck is scaling a $k$-fold adjacency matrix product to large graphs, especially the dense expected adjacency matrices.

# 6 EVALUATION

We detail our methodology and discuss our empirical results. Appendix A.5 provides training details.

## 6.1 EXPERIMENTAL DESIGN

We describe our benchmark datasets, comparison methods, and how evaluation metrics are computed.

**Datasets**  We use datasets from previous studies of GGMs (Mahmoudzadeh *et al.*, 2024; Yun *et al.*, 2019; Hao *et al.*, 2020). Cora, ACM, and CiteSeer are citation networks, IMDb is a movie dataset, and UW represents an academic department. Appendix A.1 presents dataset and preprocessing details. We report results for homogeneous versions of ACM and IMDb in the main paper, and for heterogeneous versions in Appendix A.2.

**Evaluation Metrics**  We compare rule-enhanced VGAE+R training with plain VGAE+ training, using three main metrics. In the following, we refer to a complete dataset as the **input graph**. Our evaluation measures *graph realism*—the quality of generated graphs—and the downstream task of node classification.

*Count Distance.* Given the training graph $D$, we sample one node embedding matrix $\boldsymbol{z}$ from the encoder posterior $q_\phi(\boldsymbol{z}|D)$ and then apply the decoder model eq. (5) to $\boldsymbol{z}$ to obtain the expected graph $\tilde{D}_{\boldsymbol{z}}$. We report the mean squared distance $(1/k \sum_{i=1}^{k} [n_i(D) - n_i(\tilde{D}_{\boldsymbol{z}})]^2)^{1/2}$—between the observed motif counts and the expected motif counts in the reconstructed graph—as the **count distance** (CD), where $k$ is the number of formulas.

*Graph Realism* measures how similar graphs generated by the model are to observed graphs. How to quantitatively assess generated graphs has been studied in recent papers. We adapt the SOTA approach that compares graph embeddings of the training graph to embeddings of generated graphs using Maximum Mean Distance (MMD) (O'Bray *et al.*, 2022; Thompson *et al.*, 2022; Shirzad *et al.*, 2022); see Appendix A.10 for details. The MMD metric is independent of the training objective.

**Node Classification**  To compute a node classification score, we randomly divide the nodes in the input graph into training, test and validation nodes (70%/20%/10%). The training graph is the input graph but with the test node labels removed. At test time, we run the encoder on the input graph to obtain node embeddings for all nodes, then apply the decoder to predict node labels for the test nodes.

## 6.2 EXPERIMENTAL RESULTS

*Count Distance and Graph Realism are the most important metrics for us since they directly pertain to graph generation quality.* Our graph generation baseline is the VGAE+ model trained without moment matching (i.e., $\lambda = 0$). To obtain formulas, we used the SOTA MLN structure learning system Factorbase (Qian and Schulte, 2015) with default settings (Appendix A.4). To illustrate, in the UW dataset, the learned formulas capture several patterns that express university domain knowledge, such as the following. (1) Whether a person teaches a course correlates with whether they have a

position. (2) Course teachers are more likely to be professors. (3) A person's program phase predicts their years in the program.

Table 2: Mean $\pm$ Std for Count Distance (CD↓) and Graph Realism (MMD↓) with Improvements (in scientific notation, so "e" represents $\times 10^n$).

| Dataset | Count Distance (MSE) | | | Graph Realism (MMD) | | |
|---|---|---|---|---|---|---|
| | VGAE+ | VGAE+R | Improv. (%) | VGAE+ | VGAE+R | Improv. (%) |
| Cora | $5.14e4 \pm 8.07e3$ | $2.68e4 \pm 9.79e3$ | 47.81 | $4.03e18 \pm 3.21e18$ | $6.84e17 \pm 9.15e17$ | 83.03 |
| CiteSeer | $3.40e4 \pm 1.06e3$ | $2.47e4 \pm 1.38e3$ | 27.35 | $1.20e18 \pm 3.15e17$ | $6.14e16 \pm 3.65e16$ | 94.88 |
| Computers | $4.63e5 \pm 2.67e4$ | $3.54e5 \pm 6.44e4$ | 23.61 | $3.80e25 \pm 1.86e25$ | $1.08e24 \pm 1.45e24$ | 97.16 |
| Photo | $2.01e5 \pm 1.96e4$ | $1.24e5 \pm 1.37e4$ | 38.01 | $1.16e24 \pm 8.30e23$ | $1.66e22 \pm 7.63e21$ | 98.57 |
| IMDb | $5.86e5 \pm 1.96e4$ | $3.23e5 \pm 1.34e5$ | 44.81 | $2.94e23 \pm 6.29e22$ | $4.58e22 \pm 3.25e22$ | 84.42 |
| UW | $9.92e5 \pm 8.87e4$ | $9.48e5 \pm 5.69e4$ | 4.51 | $4.72e13 \pm 1.90e13$ | $2.32e13 \pm 1.94e13$ | 50.74 |
| ACM | $1.24e5 \pm 4.86e3$ | $2.81e4 \pm 3.54e3$ | 77.34 | $3.31e20 \pm 1.32e20$ | $3.29e18 \pm 3.08e18$ | 99.01 |

### 6.2.1 COUNT DISTANCE AND GRAPH REALISM

Table 4 shows the difference between expected and observe instance counts. Both methods show large absolute distances because a VGAE model tends to produce overly dense graphs (Orbanz and Roy, 2014). However we observe a *very large improvement in the match between expected and observed counts*, at least 23% on all datasets, except for the small graph UW with an improvement of 4.51%. On the graph realism metric, Table 4 again shows large absolute distances with the training set, and *very large improvements through FO moment matching, by an order of magnitude*. Overall we conclude that unconstrained VGAE training does not match the instantiation counts of the learned formulas and that enforcing moment matching has a large impact on generated graph realism. In addition, Section A.13 shows that VGAE + R outperforms VGAE + in statistic-based MMD metrics. Also, as discussed in Appendix A.14, we report results for Count Distance Evaluation based on prior embedding sampling. Moreover, there is a report on robustness to noisy or incomplete rules for Cora dataset in Appendix A.16.

### 6.2.2 NODE CLASSIFICATION

Since SOTA performance on node classification is nearly saturated, we do not claim that VGAE+R leads to uniformly best node classification. Instead we investigate two hypotheses:

1. Rule enhancement can improve GGM-based classification when the rules capture relevant domain knowledge.

2. The VGAE+R model is competitive with current baselines.

Table 3 *shows an improvement from rule enhancement (bold)* on 4 out 7 datasets, substantive for two of them (Cora and UW). The biggest improvement is on Cora, where moment matching increase the AUC score by 10%. Even when the rules are not very relevant for the class label, moment matching decreases classification performance only slightly.

Table 6 compares the rule-enhanced VGAE+R with the recent node classification baselines, described in Appendix A.3. Our VGAE+R model shows the best node AUC classification performance on 3/6 datasets (4/6 on F1). The biggest improvement is on CiteSeer where our baselines are far from SOTA performance. GiGaMAE is a strong baseline that achieves the best result on two datasets (Table 3). Our conclusion is that *rule-enhanced graph generation supports node classification that is competitive with recent baselines*.

**Learning Curve**   We report a learning curve experiment to examine the effect of rule knowledge on data efficiency. The idea is to simulate the impact of a domain expert providing the model with a strong set of rules. We report the predictive accuracy on the test labels, after training the VGAE with and without rules on $x = 25\%, 50\%, 75\%, 100\%$ of training labels.

Figures 13 to 15 show that *moment matching improves data efficiency substantially on the CiteSeer, Cora, and Photos datasets*. For example on CiteSeer with 50% of node labels, moment matching achieves a 15% higher F1-score than baseline VGAE learning. The learning curves with and without

| Dataset | Metric | VGAE+R | VGAE+ | GiGaMAE |
|---|---|---|---|---|
| Cora | AUC | **0.965 ± 0.013** | 0.865 ± 0.043 | 0.920 |
| | F1 Score | **0.887 ± 0.016** | 0.699 ± 0.103 | 0.856 |
| UW | AUC | **0.960 ± 0.012** | 0.889 ± 0.054 | - |
| | F1 Score | **0.654 ± 0.031** | 0.618 ± 0.030 | - |
| CiteSeer | AUC | **0.903 ± 0.008** | 0.891 ± 0.013 | 0.842 |
| | F1 Score | **0.794 ± 0.042** | 0.733 ± 0.058 | 0.798 |
| Computers | AUC | 0.915 ± 0.022 | **0.920 ± 0.004** | 0.941 |
| | F1 Score | 0.827 ± 0.047 | **0.837 ± 0.005** | 0.770 |
| Photo | AUC | **0.991 ± 0.003** | 0.980 ± 0.021 | 0.963 |
| | F1 Score | **0.972 ± 0.002** | 0.946 ± 0.052 | 0.569 |
| ACM | AUC | 0.761 ± 0.077 | **0.775 ± 0.074** | 0.823 |
| | F1 Score | **0.525 ± 0.014** | 0.523 ± 0.009 | 0.440 |
| IMDb | AUC | 0.829 ± 0.006 | **0.828 ± 0.011** | 0.890 |
| | F1 Score | **0.697 ± 0.008** | 0.687 ± 0.014 | 0.457 |

Table 3: Node classification results for graph generation with and without rule enhancement. The recent GiGAMAE system is a strong baseline. Bold indicates the best VGAE score, underline the best GiGAMAE score. Standard deviations are reported for five random weight initializations.

moment matching are similar for the datasets ACM, IMDb and Computers because their rules affect node classification little.

**Impact of Rules on Training** Figure 19 shows the node label loss component of decoder training Equation (2) for the CiteSeer dataset. Rule matching adds a difficult new component to the VGAE+ objective, which initially causes a spike in the label loss component. After the VGAE+R model has encoded the background knowledge in its weights, it learns to optimize the other components, including the node label loss. This shows that *rule matching is a strong regularizer* that takes the network to a very different part of weight space compared to the baseline VGAE+ loss, and supports better generalization. The supplement Appendix A.12 illustrates this pattern in loss curves.

## 7 CONCLUSION, LIMITATIONS AND FUTURE WORK

We proposed a new semantic loss objective function for training a deep graph generative model (GGM) to incorporate FO domain knowledge expressed by logical formulas: Maximize the data likelihood subject to a moment matching constraint, which requires *the expected formula instance counts under a model to match the observed instance count.* Our main algorithmic contribution is a *new differentiable matrix multiplication method for computing both observed and expected counts.* In empirical evaluation, we found that moment matching improves the quality of the graphs generated by a Variational Graph Auto-Encoder (VGAE) model by an order of magnitude or more, both with respect to instance counts and with respect to a standard metric of graph realism. Applying the trained GGM to the downstream task of node classification, moment matching improved classification accuracy on all but one of our benchmark datasets. The domain knowledge incorporated in the model is often effective in improving predictions from small datasets, as shown in learning curves.

*Limitations.* As our paper is the first to combine deep graph generation with a first-order semantic loss, it leaves several aspects open for future developments. (1) Scaling the matrix multiplication algorithm for expected counts is a challenge (Section 5). There is a report on scalability and runtime analysis in Appendix A.15. A possible solution are approximation algorithms from the related problem of weighted model counting (van Bremen and Kuzelka, 2020). (2) An incorrect or incomplete set of rules limits the effectiveness of the semantic loss function. We did not explore methods for validating the knowledge expressed in formulas, such as human-in-the-loop. (3) Because rule learners (MLN structure learners) assume a single dataset, we did not explore enhancing GGMs other than VGAEs (e.g., auto-regressive, diffusion, and matching flow models (cf. Section 2).

In sum, moment matching presents a novel semantic loss approach to neuro-symbolic AI that combines logical rules with deep graph learning. Our experiments show great potential for enhancing deep graph generative models with rule-based knowledge.

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

# A  APPENDIX / SUPPLEMENTAL MATERIAL

## A.1  DATASET INFORMATION

To evaluate all the methods we utilize 5 datasets used in previous studies of generative models Kipf and Welling (2016); Yun *et al.* (2019); Hao *et al.* (2020).

- **Cora** (Kipf and Welling, 2016) is a citation dataset that consists of nodes that represent machine-learning papers divided into seven classes (the subjects of the papers) and links that represent citation between them. The target for node classification is the subject of the papers. This dataset has 5,429 links, 2,708 nodes with an average node degree of 3.8.

- **ACM** (Yun *et al.*, 2019) is a citation dataset. It has three types of nodes (paper, author, and venue) and four types of links. The target for node classification is predicting one of three labels corresponding to the conferences where the papers were published. This dataset has 18,929 links, 8,993 nodes with an average node degree of 2.209.

- **IMDb** (Yun *et al.*, 2019) is a movie dataset with three types of nodes (movies, actors, and directors) and it uses the genre of movies as their labels. The target for node classification is predicting one of three genres of movies. This dataset has 19,120 links, 12,772 nodes with an average node degree of 2.9.

- **CiteSeer** (Kipf and Welling, 2016) is also a citation dataset that consists of nodes that represent machine-learning papers divided into six classes (the topics of the papers) and links that represent citation between them. The target for node classification is the topic of the publications. This dataset has 4,732 links, 3,327 nodes with an average node degree of 2.7.

- The **UW** dataset models academic relationships at a university, where persons can be both students and professors and key relations include AdvisedBy (a student advised by a professor) and TaughtBy (a student taking a course taught by a professor). The dataset contains 278 person entities and 132 course entities. The classification target is the phase of study of the student, which can be one of three labels: *pre-quals*, *post-quals*, or *post-generals*. If the person is not a student, the label is 0.

- **Photo & Computers** are datasets from the Amazon co-purchase graph (McAuley *et al.*, 2015). In these datasets, nodes represent goods, links indicate that two goods are frequently bought together, node features are bag-of-words encoded product reviews, and class labels are given by the product category (Hao *et al.*, 2020).

**Data Preprocessing**   Following previous work (Kipf and Welling, 2016), for GNN message passing we add self-loops and make all links undirected (i.e., if the training data contains an adjacency, $v \to u$, it also contains $u \to v$.) Cora and CiteSeer are homogeneous datasets, whereas ACM and IMDb are heterogeneous datasets. Rule learning is applied to the original data.

## A.2  RESULTS FOR HETEROGENEOUS ACM AND IMDB

We present graph generation results for the original heterogeneous versions of IMDb and ACM, which feature different types of nodes and links. For ease of comparison, we repeat the results for homogeneous versions from the main paper. Heterogeneous versions are distinguished by an asterisk. We observe large percentage improvements from rule enhancement, especially for IMDb.

Table 4: Mean $\pm$ Std for Count Distance (CD$\downarrow$) and Graph Realism (MMD$\downarrow$) with Improvements (in scientific notation, so "e" represents $\times 10^n$).

| Dataset | Count Distance (MSE) | | | Graph Realism (MMD) | | |
|---|---|---|---|---|---|---|
| | VGAE+ | VGAE+R | Improv. (%) | VGAE+ | VGAE+R | Improv. (%) |
| ACM | $1.24e5 \pm 4.86e3$ | $2.81e4 \pm 3.54e3$ | 77.34 | $3.31e20 \pm 1.32e20$ | $3.29e18 \pm 3.08e18$ | 99.01 |
| ACM* | $3.65e5 \pm 3.32e3$ | $1.47e4 \pm 2.54e3$ | 95.97 | $1.9e15 \pm 1.32e15$ | $9.8e15 \pm 0.98e15$ | $-415.79$ |
| IMDb | $5.86e5 \pm 1.96e4$ | $3.23e5 \pm 1.34e5$ | 44.81 | $2.94e23 \pm 6.29e22$ | $4.58e22 \pm 3.25e22$ | 84.42 |
| IMDb* | $1.37e5 \pm 2.23e2$ | $1.43e4 \pm 1.98e2$ | 89.56 | $1.9e20 \pm 2.88e18$ | $7.8e17 \pm 1.32e17$ | 99.59 |

For node classification on heterogeneous datasets, the class labels are as follows: For IMDb*, predict the genre of the movie nodes, and for ACM*, predict the area of a paper. As with the homogeneous versions, rule enhancement does not lead to substantive improvements on these datasets, because the rules are not informative for node classification in these datasets.

Table 5: AUC and F1 Score for link prediction on heterogeneous ACM and IMDb (higher is better).

| Dataset | Model | AUC | F1 Score |
|---------|-------|-----|----------|
| ACM* | VGAE+R | 0.976 ± 0.004 | **0.9184 ± 0.011** |
| | VGAE+ | **0.979 ± 0.003** | 0.9123 ± 0.011 |
| IMDb* | **VGAE+R** | 0.774 ± 0.006 | **0.632 ± 0.008** |
| | VGAE+ | **0.769 ± 0.011** | 0.615 ± 0.014 |

### A.3 NODE CLASSIFICATION BASELINE METHODS

Table 6 compares the rule-enhanced VGAE+R with the following node classification baselines. We focused on baselines that are similar to generative models in that they aim to support multiple prediction tasks. The baselines apply to homogeneous graphs only, so we homogenized ACM and IMDb (Mahmoudzadeh *et al.*, 2024) and omitted UW.

**Generalizable Graph Masked AutoEncoder (GiGaMAE)** (Shi *et al.*, 2023) introduces a novel graph encoder based on aligning different node embeddings that respectively encode structural and feature information. **MVGRL** is an inductive self-supervised approach for learning representations of nodes and graphs by contrasting different structural views of graphs (Hassani and Khasahmadi, 2020). $\mathcal{G}^2\text{Pxy}$ (Zhang *et al.*, 2023) generates proxy nodes to support classification. For all methods we used the authors' code to train a node classifier and generate class labels.

| Dataset | Model | AUC | F1-score |
|---------|-------|-----|----------|
| Cora | VGAE+R | **0.965 ± 0.013** | **0.887 ± 0.016** |
| | GiGaMAE | 0.920 | 0.856 |
| | $\mathcal{G}^2 P_{xy}$ | 0.921 | 0.781 |
| | MVGRL | 0.888 | 0.886 |
| CiteSeer | VGAE+R | **0.903 ± 0.008** | 0.794 ± 0.042 |
| | GiGaMAE | 0.842 | **0.798** |
| | $\mathcal{G}^2 P_{xy}$ | 0.850 | 0.781 |
| | MVGRL | 0.807 | 0.710 |
| Computers | VGAE+R | 0.915 ± 0.022 | **0.827 ± 0.047** |
| | GiGaMAE | 0.941 | 0.770 |
| | $\mathcal{G}^2 P_{xy}$ | 0.680 | 0.578 |
| | MVGRL | **0.983** | 0.816 |
| Photo | VGAE+R | **0.991 ± 0.003** | **0.972 ± 0.002** |
| | GiGaMAE | 0.963 | 0.569 |
| | $\mathcal{G}^2 P_{xy}$ | 0.773 | 0.513 |
| | MVGRL | 0.963 | 0.960 |
| ACM | VGAE+R | 0.761 ± 0.084 | 0.525 ± 0.014 |
| | GiGaMAE | **0.823** | 0.440 |
| | $\mathcal{G}^2 P_{xy}$ | 0.742 | 0.661 |
| | MVGRL | 0.708 | **0.803** |
| IMDb | VGAE+R | 0.829 ± 0.006 | **0.697 ± 0.008** |
| | GiGaMAE | **0.890** | 0.457 |
| | $\mathcal{G}^2 P_{xy}$ | 0.654 | 0.541 |
| | MVGRL | 0.788 | 0.653 |

Table 6: Node classification results comparing rule-enhanced graph generation against baselines.

## A.4 FORMULA LEARNING

In principle, any Markov Logic Network structure learning method can be used with our moment-matching training objective. We deployed the Factorbase sytem (Qian and Schulte, 2015) for several reasons.

1. Interpretability: The first phase of Factorbase uses Bayesian network structure learning method to discover a *causal graph* that provides a perspicious visual representation of all associations learned from the data.

2. Scalability: Factorbase scales to larger and more complex datasets than other MLN structure learning method, on the order of 100K nodes.

3. Quality: Research has shown that the formulas discovered by Factorbase support high quality inferences in relational prediction tasks.

Factorbase has three main phases.

1. Causal Graph Discovery.

2. Rule Extraction from the learned causal graph.

3. Moralization: convert learned rules to conjunctions.

Both rule extraction and moralization are computationally straightforward linear-time operations. Moralization is a method for converting learned rules to conjunctions. Previous research in statistical-relational learning (Kazemi *et al.*, 2014; Khosravi *et al.*, 2012) has shown that moralization is a strong method for converting learned rules to useful features for generative graph models. Note that therefore *our matrix multiplication algorithm can be applied to any set of learned rules after moralization*. This means that our algorithm for rule enhancement can be applied to any set of learned rules, not only those obtained from MLN structure learning. We next explain in the phases of Factorbase formula discovery and illustrate the results on our dataset.

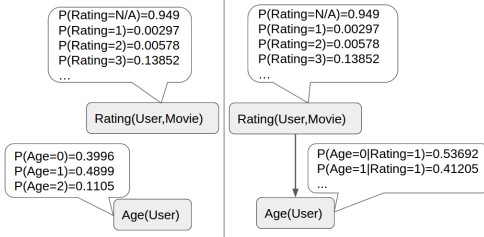

Figure 5: Example First-Order Causal Graphs: left = $B_1$ with graph $\mathcal{G}_1$, right = $B_1^+$ with graph $\mathcal{G}_1^+$.

**Causal Graph Discovery** A **causal graph (CG) structure** is a directed acyclic graph $G$ (DAG) whose nodes comprise a set of random variables Pearl (2000). The interpretation is that the parents of a node represent its direct causes Pearl (2000). A **causal Bayesian network** $B$ is a structure $G$ together with a set of parameter values of the form $P(child\_value|parent\_values)$, which specify the distribution of a child node given an assignment of values to its parent node. A first-order causal graph Wang *et al.* (2008); Kimmig *et al.* (2014) (FOCG) is a CG whose nodes are first-order terms. The CG parameters specify the distribution of a child node given an assignment of values to its parent node. Figure 5 shows two parametrized FOCGs. The right FOCG connects the rating of a movie to the age of the rater. In the left FOCG, ratings are independent of ages. The process of structure learning searches for statistically significant connections between first-order terms to introduce or remove edges (Schulte and Qian, 2018; Schulte and Gholami, 2017).

We show the causal graphs learned by Factorbase for three datasets. We find that many of the learned rules capture intuitively plausible domain constraints, such as homophily: if one paper cites another, then they are likely from the same research area. For example in the graph for the Cora dataset, the parents of the target label of a paper (node) are paper features 1 and 2, the citation relationship between a paper and the target paper (denoted "edges_table"), and the label of the citing paper. The

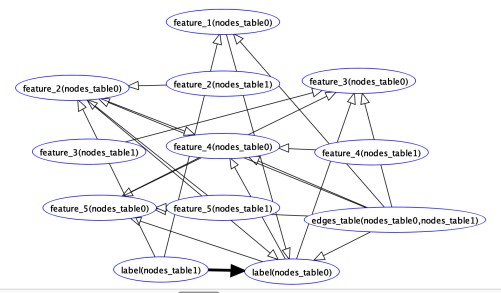

Figure 6: The causal graph for the Cora dataset. We have highlighted the homophily edge that shows that if paper 0 cites paper 1, then the label of paper 0 correlates with the label of paper 1.

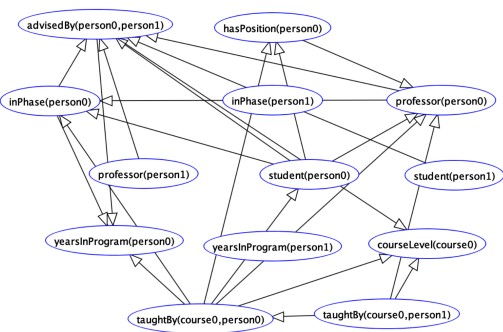

Figure 7: The causal graph for the UW dataset.

graph defines a rule for each combination of the 4 parent nodes, and each of the 7 possible child node values (paper classes).

ACM and CiteSeer are citation networks like Cora with a similar graph, which we omit. Figure 7 shows the causal graph for the UW dataset. The graph captures a number of relationships that make sense in the university domain. For example whether a person teaches a course correlates with whether they have a position.

Figure 8 shows the causal graph for the IMDb dataset. The graph captures many correlations between features of movies and actors. An interesting aspect of the graph is that for the target label (genre of the movie), its only parents are other features of the movie to be classified (features 1,2,4). This suggests that relational features, such as the features of actors who appear in the movie, should not add information beyond movie features. We verified this directly by building a classifier based on movie features only, which performed better than the GNN that uses relational features.

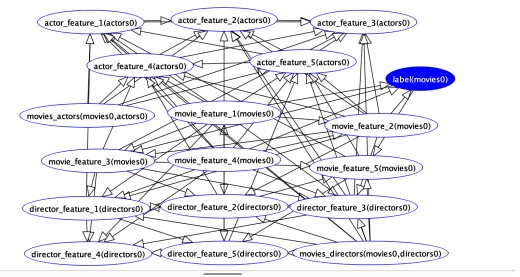

Figure 8: The causal graph for the IMDb dataaset.

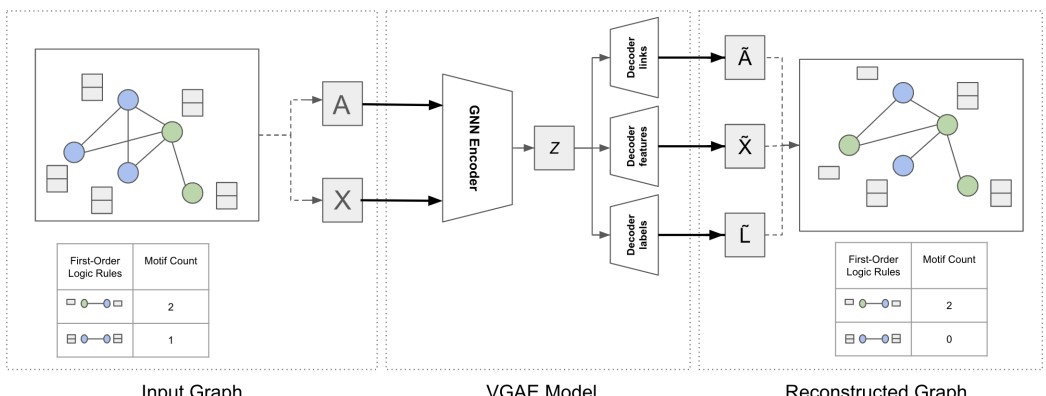

Figure 9: The FO conjunction $Age(User) = 1$, $Rating(User, Movie) = 4$ can be viewed as a two-node motif. The conjunction is obtained from the rule $Age(User) = 1 \rightarrow Rating(User, Movie) = 4$ by moralization.

Figure 10: Encoder-Decoder Training Architecture for the VGAE+ Model.

**Rule Extraction from Causal Graphs**   A **rule** is of the form $B \rightarrow H$ where the body $B$ is an FO conjunction and the head $H$ is a single FO literal. An example rule is

$$Age(User) = 1 \rightarrow Rating(User, Movie) = 4.$$

This rule expresses the knowledge that the age level of a user influences their ratings.

A single rule $B \rightarrow H$ is defined by a combination $parent\_values \rightarrow child\_value$, for each child node value and possible combination of parents values (Khosravi *et al.*, 2012). For example, the causal graph of Figure 5(right) defines the rule

$$Rating(User, Movie) = 3 \rightarrow Age(User) = 1).$$

Given 6 possible rating values and 5 age levels, the link $Rating(User, Movie) \rightarrow Age(User))$ defines $6 \times 5 = 30$ rules.

As the Factorbase system can output many rules, we use $2n(B, H) \ln(n(H|B)/n(H)) - \ln n(H)$ as a *rule quality metric* for pruning. The conditional count is given by $n(H|B) = n(H, B)/n(B)$. This metric computes the increase in the log-probability of the head given the body, relative to the prior probability of the head, together with a BIC-type correction for sample size (Schulte and Gholami, 2017). The quality metric can be interpreted as a positive local BIC model selection score. We keep all rules whose quality metric is above 0. Rule pruning reduces the number of rules for scalability, and also simulates the impact of a domain expert selecting the most important rules.

**Moralization: From rules to conjunctions**   Most rule learners are based on *discriminative* learning, building on classification techniques to search for bodies that predict the head. A question researched in SRL is how to convert a set of rules to a set of graph features/statistics that support *generative* graph modelling. The recommended answer is *moralization*: convert each rule to a conjunction $\phi = (B, H)$ (Domingos and Richardson, 2007, 12.5.3), (Kazemi *et al.*, 2014). Figure 9 illustrates how moralization converts a rule to a conjunction, and how a conjunction represents a motif.

A.5   DETAILS ON THE VGAE+R MODEL

Figure 10 shows the VGAE+R architecture. The graph **encoder** $q_\phi(z|X, A)$ is implemented by a GNN that takes as input an attribute graph and returns latent node embeddings. For compatibility with

baseline methods, the encoder does not receive node labels as input, but adding them is straightforward. Following Mahmoudzadeh *et al.* (2024), we used a multi-layer graph attention network (GAT) and an RGCN (Schlichtkrull *et al.*, 2018) for heterogeneous datasets. The encoders were configured in the same way as Mahmoudzadeh *et al.* (2024).

The decoders generate links, node features, and node labels independently as follows.

$$p_\eta(\mathbf{A}|\boldsymbol{z}) = \prod_{r=1}^{T} \prod_{u,v} p_{\eta_r}(\mathbf{A}_r[u,v]|\boldsymbol{z}[u], \boldsymbol{z}[v]) \tag{5}$$

$$p_\psi(\boldsymbol{X}|\boldsymbol{z}) = \prod_u p_\psi(\boldsymbol{X}[u]|\boldsymbol{z}[u]) \tag{6}$$

$$p_\phi(\boldsymbol{L}|\boldsymbol{z}) = \prod_u p_\phi(\boldsymbol{L}[u]|\boldsymbol{z}[u]), \tag{7}$$

Jaeger and Schulte (2020) provide a strong theoretical foundation for the independent decoder model. Following Mahmoudzadeh *et al.* (2024), we use a stochastic blocks model for the link decoder Equation (5), and MLPs for the feature and label decoders Equations (6) and (7). When training a model for graph generation, we use the training objective Equation (2), without the node labels. When training a model for node classification, the encoder does not see the node labels, but the decoder has access to the training node labels to evaluate the node label term Equation (7). At test time, the trained encoder is applied to the available graph information to produce node embeddings for each test node, and then the trained decoder is applied to output class probabilities.

**Hyperparameters** We used the ADAM optimizer for training, with default hyperparameter values. For *graph generation* we obtained the best results for both the VGAE+ and the VGAE+R model by setting all ELBO hyperparameters $\alpha, \beta, \gamma$ to 1, similar to Simonovsky and Komodakis (2018). The motif loss weight $\lambda$ was also set to 1 except for Computers, where $\lambda = 2$ yielded better results. Both models were trained for 100 epochs on all datasets.

For *node classification*, for the VGAE+ baseline model with $\lambda = 0$, we follow Mahmoudzadeh *et al.* (2024) and use Bayesian optimization to fix the ELBO hyperparameters $\alpha, \beta, \gamma$. For the VGAE+R model, after fixing the ELBO hyperparameters, we set the motif loss weight $\lambda$ empirically based on a validation set. We found that different datasets require different training regimes to build good node classifiers. Specifically, the number of training epochs were as follows for both VGAE+ and VGAE+R models: 700 for Cora and CiteSeer, 1000 for IMDb, Computers, ACM, Photo, and 100 for UW.

For the learning curves, we trained the model for 1000 epochs at each sample size on the Citeseer dataset. To obtain standard deviations, we initialized training with five different seeds for node classification, and three different seeds for graph generation.

**Computing Resources** The experiments were conducted using a GPU cluster. The compute resources varied based on dataset size. For large datasets such as Computers and Photos, experiments were run on NVIDIA A100 GPUs with 80GB VRAM, 64GB RAM, and 16 CPU threads. For IMDB and ACM, NVIDIA A40 GPUs with 48GB VRAM, 64GB RAM, and 16 CPU threads were used. For smaller datasets such as Cora, Citeseer, and UW, experiments were conducted on NVIDIA GeForce GTX 1080 Ti GPUs. Each dataset required approximately 20GB of storage for generated graphs. Given the training regime described, generating measurements for one dataset took approximately 4 to 5 hours.

## A.6 First-order Semantic Loss vs. Propositional Semantic Loss

In this section we show how the semantic loss Equation (4) with the distance function

$$|\ln n_i(D) - \ln E_{\boldsymbol{\theta}}[n_i|\boldsymbol{z}]| \tag{8}$$

reduces to the semantic loss function of Xu *et al.* (2018) in the case of a propositional formula $\phi_i$. For a given graph, a propositional formula is either true or false in the graph. Xu *et al.* give the

example of an "exactly-one" constraint that says that each node is assigned exactly to exactly one class. Translating into our notation, a propositional formula has an instance count of 1 if it is satisfied in a graph, and 0 otherwise:

$$n_i(\mathcal{G}) \leq 1 \text{ for propositional } \phi_i \tag{9}$$

The basis idea of Xu *et al.* is to view a propositional formula as a constraint that a model should satisfy. There the semantic loss function should maximize the probability of the formula under the model. The probability of the formula is the probability of the graphs that satisfy it. The steps in the proof are as follows.

$$p_{\boldsymbol{\theta}}(\phi_i|\boldsymbol{z}) = \sum_{\mathcal{G}} P_{\boldsymbol{\theta}}(\mathcal{G}|\boldsymbol{z})n_i(\mathcal{G}) = E_{\boldsymbol{\theta}}[n_i|\boldsymbol{z}] \tag{10}$$

$$-\ln p_{\boldsymbol{\theta}}(\phi_i|\boldsymbol{z}) = -\ln E_{\boldsymbol{\theta}}[n_i|\boldsymbol{z}] \tag{11}$$

$$|\ln n_i(D) - \ln E_{\boldsymbol{\theta}}[n_i|\boldsymbol{z}]| = |0 - \ln E_{\boldsymbol{\theta}}[n_i|\boldsymbol{z}]| = |-\ln p_{\boldsymbol{\theta}}(\phi_i|\boldsymbol{z})| = -\ln p_{\boldsymbol{\theta}}(\phi_i|\boldsymbol{z}) \tag{12}$$

Equation (10) holds because due to the zero-one condition (9) the probability of a formula is the same as the expected instance count, which immediately implies Equation (11). For Equation (12), since formula $\phi_i$ is assumed true in *all* graphs, in particular it will be true in the observed graph $D$. Therefore $n_i(D) = 1$. The last equality holds because for $p_{\boldsymbol{\theta}} \leq 0$, we have $-\ln p_{\boldsymbol{\theta}}(\phi_i|\boldsymbol{z}) \geq 0$, so $-\ln p_{\boldsymbol{\theta}}(\phi_i|\boldsymbol{z}) = |-\ln p_{\boldsymbol{\theta}}(\phi_i|\boldsymbol{z})|$. The leftmost term in Equation (12) is our first-order semantic loss, and the rightmost is the propositional first-order semantic loss of Xu *et al.*, which establishes our claim.

### A.7 Expanded First-Order Logic Definitions

We add to the definitions in the main body to introduce concepts we need in our proofs and in the full description of matrix multiplication algorithm. We make this section self-contained for ease of reference. A positive **relationship literal** is of the form $R(U, V)$. A **negative** relationship literal is of the form $\neg R(U, V)$. A generic relationship literal (positive or negative) is denoted as $\ell(U, V)$. A **ground** relationship literal is of the form $\ell(U = u, V = v)$ where $u$ and $v$ are two node indices. Similar to specifying arguments for variables in a programming language, grounding specifies arguments for node variables. Positive and negative unary literals are defined similarly.

For a positive ground relationship literal $\ell(U = u, V = v) = R(U = u, V = v)$, the indicator $I_{\mathcal{G}}(\ell(U = u, V = v)) = 1$ if the two nodes $u$ and $v$ are linked by edge type $R$ in graph $G$. For a negative ground relationship literal, $\ell(U = u, V = v) = \neg R(U = u, V = v)$, we have $I_{\mathcal{G}}(\neg R(U = u, V = v)) = 1 - I_{\mathcal{G}}(R(U = u, V = v))$.

A conjunction is a list of literals. Intuitively, a conjunction is a template for a motif or frequently occurring subgraph. The indicator function specifies which nodes satisfy the literal/conjunction. Our formal definitions are as follows.

A **conjunction** $\phi$ comprises three elements:

1. A list $\ell_1(U_1, V_1), \ldots, \ell_P(U_P, V_P)$ comprising $P$ relationship literals where $U_i$ and $V_i$ are node variables.

2. A list $\ell_1(W_1), \ldots, \ell_Q(W_Q)$ comprising $Q$ unary literals, where $W_i$ is a node variable.

3. A set of equality constraints $EQ$ of the form $D_k = E_k$ for any two node variables $D_k$ and $E_k$ that appear in the list of relationship literals or unary literals. Formally, $EQ$ is a set of unordered pairs of node variables.

This definition is equivalent to the definition in the main paper: all node variables are assumed to occur exactly once in a conjunction, and the equality constraints specify which node variables must be mapped to the same node indices. Representing equality constraints in an explicit list facilitates the statement of our matrix multiplication algorithm.

A conjunction with $P$ relationship literals $\ell_1(U_1, V_1), \ldots, \ell_P(U_P, V_P)$ is a **chain conjunction** if there is a permutation $\pi$ of the literals such that the equality constraints comprise $V_{\pi(i-1)} = U_{\pi(i)}$

for $i = 2, \ldots, P$. A conjunction $\phi$ has $2P + Q$ parameters (i.e., node variables). Specifying a list of $2P + Q$ node indices as arguments to the conjunction returns a **grounded conjunction.** A grounded chain conjunction corresponds to a path in the graph where each consecutive pair of nodes is connected by an edge.

The indicator function for a grounded conjunction is given by:

$$
I_{\mathcal{G}}(\ell_1(U_1 = u_1, V_1 = v_1), \ldots, \ell_P(U_P = u_P, V_P = v_P),
$$
$$
\ell_1(W_1 = w_1), \ldots, \ell_Q(W_Q = w_Q))
$$
$$
= \prod_{i=1}^{P} I_{\mathcal{G}}(\ell_i(U_i = u_i, V_i = v_i)) \prod_{j=1}^{Q} I_{\mathcal{G}}(\ell_j(W_j = w_j))
$$

**Conjunction Counts.** For compactness, write a grounding as $\langle \boldsymbol{U} = \boldsymbol{u}, \boldsymbol{V} = \boldsymbol{v}, \boldsymbol{W} = \boldsymbol{w} \rangle$ so the indicator function returns a 0/1 value for $I_{\mathcal{G}}(\langle \boldsymbol{U} = \boldsymbol{u}, \boldsymbol{V} = \boldsymbol{v}, \boldsymbol{W} = \boldsymbol{w} \rangle, EQ)$. A grounding $\langle \boldsymbol{U} = \boldsymbol{u}, \boldsymbol{V} = \boldsymbol{v}, \boldsymbol{W} = \boldsymbol{w} \rangle$ is **valid** for a set of equality constraints $EQ$ if for any two assignments $D = d$ and $E = e$ we have $d = e$ if and only if $(D = E) \in EQ$. Thus node variables constrained to be equal must be assigned the same node, and otherwise must be assigned different nodes. We write $Valid_{EQ}$ for the set of valid groundings. The **instance count** for a conjunction $\phi$ in a graph $\mathcal{G}$ returns the number of valid groundings that satisfy the conjunction:

$$
n_\phi(\mathcal{G}) = \sum_{\langle \boldsymbol{u}, \boldsymbol{v}, \boldsymbol{w} \rangle \in Valid_{EQ}} I_{\mathcal{G}}(\langle \boldsymbol{U} = \boldsymbol{u}, \boldsymbol{V} = \boldsymbol{v}, \boldsymbol{W} = \boldsymbol{w} \rangle)
$$

Here, we evaluate the indicator function for each combination of $\langle \boldsymbol{u}, \boldsymbol{v}, \boldsymbol{w} \rangle$, and sum up the values for all combinations to obtain the desired count of satisfying groundings.

### A.8  PROOF OF PROPOSITION  2

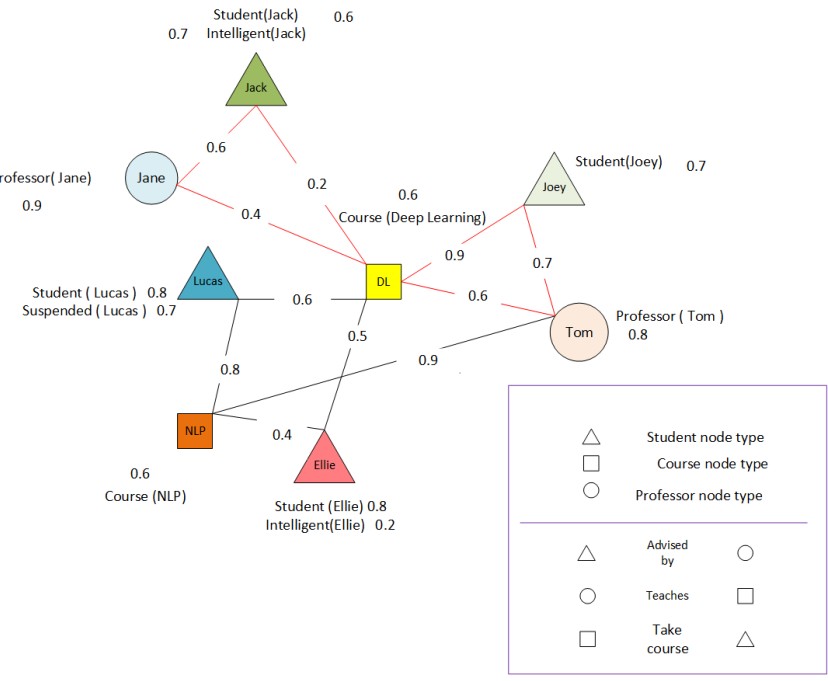

Figure 11: An example expected graph corresponding to Figure 4.

The output of the VGAE+ model is an expected graph like Figure 11. The following proposition states that the expected counts can be computed based on a expected graph. We use the full first-order logic definitions of Appendix A.7.

**Proposition 2.** *The expected conjunction count given a set of node embeddings can be computed as the conjunction count in the expected graph:* $E_{\mathcal{G} \sim P(\mathcal{G}|\boldsymbol{z})}[n_\phi(\mathcal{G})] = n_\phi(\tilde{\mathcal{G}}_{\boldsymbol{z}})$

*Proof.* Without loss of generality, assume the conjunction is of the form $\ell_1(U_1 = u_1, V_1 = v_1), \ldots, \ell_P(U_P = u_P, V_P = v_P), \ell_1(W_1 = w_1), \ldots, \ell_Q(W_Q = w_Q)$ with equality constraints $EQ$. Define the random variables $l_i^{uv}, i = 1, \ldots, P$ to return the indicator value $I_{\mathcal{G}}(\ell_i(U_i = u, V_i = v))$ and $l_j^w, j = 1, \ldots, Q$ to return the indicator value $I_{\mathcal{G}}(\ell_j(W_j = w))$ . Then

$$
\begin{aligned}
&E_{\mathcal{G} \sim P(\mathcal{G}|\boldsymbol{z})}[n_\phi(\mathcal{G})] \\
&= E[\sum_{\langle \boldsymbol{u}, \boldsymbol{v}, \boldsymbol{w} \rangle \in Valid_{EQ}} \prod_{i=1}^{P} I_{\mathcal{G}}(\ell_i(U_i = u_i, V_i = v_i)) \\
&\quad \prod_{j=1}^{Q} I_{\mathcal{G}}(\ell_j(W_j = w_j))] \\
&= E[\sum_{\langle \boldsymbol{u}, \boldsymbol{v}, \boldsymbol{w} \rangle \in Valid_{EQ}} \prod_{i=1}^{P} l_i^{\boldsymbol{u}_i \boldsymbol{v}_i} \prod_{j=1}^{Q} l_j^{\boldsymbol{w}_j}] \\
&= \sum_{\langle \boldsymbol{u}, \boldsymbol{v}, \boldsymbol{w} \rangle \in Valid_{EQ}} E[\prod_{i=1}^{P} l_i^{\boldsymbol{u}_i \boldsymbol{v}_i} \prod_{j=1}^{Q} l_j^{\boldsymbol{w}_j}] \\
&= \sum_{\langle \boldsymbol{u}, \boldsymbol{v}, \boldsymbol{w} \rangle \in Valid_{EQ}} \prod_{i=1}^{P} E[l_i^{\boldsymbol{u}_i \boldsymbol{v}_i}] \prod_{j=1}^{Q} E[l_j^{\boldsymbol{w}_j}] \\
&= \sum_{\langle \boldsymbol{u}, \boldsymbol{v}, \boldsymbol{w} \rangle \in Valid_{EQ}} \prod_{i=1}^{P} p_{\tilde{\mathcal{G}}_{\boldsymbol{z}}}(\ell_i(U_i = \boldsymbol{u}_i, V_i = \boldsymbol{v}_i)) \\
&\quad \prod_{j=1}^{Q} p_{\tilde{\mathcal{G}}_{\boldsymbol{z}}}(\ell_j(W_j = \boldsymbol{w}_j)) = n_\phi(\tilde{\mathcal{G}}_{\boldsymbol{z}})
\end{aligned}
\tag{13}
$$

Equation (13) follows because the expectation of a product of independent random variables is the product of their expectations. The random variables $l_i^{\boldsymbol{u}_i \boldsymbol{v}_i}$ and $l_j^{\boldsymbol{w}_j}$ are independent because the (in)equality constraints ensure that in a valid grounding, no two different literals are ground to the same ground literal. And conditional on the node embeddings $\boldsymbol{z}$, any two different ground literals are independent. $\qquad \square$

### A.9 MATRIX MULTIPLICATION METHOD

#### A.9.1 EXAMPLE FOR EXPECTED INSTANCE COUNT

Figure 12 illustrates the matrix multiplication method for the expected graph in our example.

Figure 12: The matrix multiplication sequence for our example conjunction and Figure 11 graph.

### A.9.2 FULL SPECIFICATION OF MATRIX MULTIPLICATION METHOD

Intuitively, a chain conjunction is a template for a motif or frequently occurring subgraph. The indicator function specifies which nodes satisfy the literal/conjunction. Our formal definitions are as follows.

The input to our counting algorithm is a chain conjunction $\{\ell_1(U_1, V_1), \ldots, \ell_P(U_P, V_P), \ell_1(W_1), \ldots, \ell_Q(W_Q), EQ\}$ and an input graph $\mathcal{G}$. The first step is to process the unary literals by masking adjacency matrix entries of nodes that do not satisfy all unary literals. The second step is to define inductively a sequence of matrices such that the instance count of the conjunction can be computed as the entry sum of the matrix product. We use $A \circ B$ to denote the element-wise matrix (Hadamard) product and $I$ for the identity matrix of the appropriate dimension. A positive relationship literal $R(U, V)$ is **associated with** $\mathbf{A}_r$, the adjacency matrix for relation $R$. A negative relationship literal $\neg R(U, V)$ is associated with $\neg \mathbf{A}_r$ where $\neg \mathbf{A}_r[u, v] = 1 - \mathbf{A}_r[u, v]$ for all node indices $u, v$.

**Step 1: Unary Literals**   Consider binary relationship literal $\ell_i(U, V)$ with associated $m \times n$ adjacency matrix $A_i(\mathcal{G})$. We search for every unary literal $\ell(W)$ where $(U = W) \in EQ$. For each such literal, we create a binary vector $T$ of size $m$ such that $T[w] := I_\mathcal{G}(\ell(W = w))$. Thus the entry $T[w]$ masks all the nodes that do not satisfy the unary literal. We apply the mask to the $w$ row of matrix $A_i$, setting $\bar{\mathbf{A}}_i[w, :] = A_i[w, :] \circ T[w]$. where $A(w, :)$ represents the entire $w$ row of matrix A. If the $w$ entry of $T$ is zero, the entire row of $\bar{\mathbf{A}}_i$ is set to 0. If the $w$ entry of $T$ is one, the entire row of $A_i$ is copied to $\bar{\mathbf{A}}_i$.

Similarly, if a unary literal $\ell(W)$ exists where $(V = W) \in EQ$, we mask the corresponding column entries in the adjacency matrix $A_i$, and repeat the masking process for all such unary literals. We refer to the adjacency matrix that incorporates the unary functor constraints as the *masked adjacency matrix* $\bar{\mathbf{A}}_i$.

**Step 2: Binary Literals**   A chain conjunction is centered if all equality constraints for the binary literals (other than the chain constraints) involve the first node variable, that is they are of the form $U_1 = E_k$. For a centered chain, we define a sequence of matrix multiplications as follows.

1. For a single literal conjunct $\phi = \ell(U, V)$ with associated masked matrix $\bar{\mathbf{A}}$, let

$$O_1(\phi) = \begin{cases} \bar{\mathbf{A}}, & \text{if } U = V \notin EQ \\ \bar{\mathbf{A}} \circ I, & \text{if } U = V \in EQ \end{cases}$$

   $\bar{\mathbf{A}} \circ I$ agrees with $\bar{\mathbf{A}}$ on the diagonal and is 0 off-diagonal.

2. Inductively, consider a conjunction $\phi$ of length k+1 in the form of $\phi = \phi', \ell_{k+1}(U_{k+1}, V_{k+1})$ where $\phi' = \ell_1(U_1, V_1), \ldots, \ell_k(U_k, V_k)$ is a conjunction of length $k$. Let

$$O_{k+1}(\phi) = \begin{cases} O_k(\phi')\bar{\mathbf{A}}_{k+1}, & \text{if } U_1 = V_{k+1} \notin EQ \\ (O_k(\phi')\bar{\mathbf{A}}_{k+1}) \circ I, & \text{if } U_1 = V_{k+1} \in EQ \end{cases} \tag{14}$$

### A.9.3 CORRECTNESS PROOF (PROPOSITION 3)

We next formulate the proposition that for every chain conjunction, there is a corresponding sequence of matrix multiplication operations such that: for every input graph $\mathcal{G}$, applying the operation sequence to the graph edge label tensor returns the instance count. In this formulation, we use the following facts: 1) Every ground positive (negative) relationship literal corresponds to a link present (absent) in the graph. 2) A grounded chain conjunction corresponds to a path in the graph where each consecutive pair of nodes is connected by a present/absent link.

**Proposition 3.** *Let* $\{\ell_1(U_1, V_1), \ldots, \ell_P(U_k, V_k), \ell_1(W_1), \ldots, \ell_Q(W_Q), EQ\}$ *be a chain conjunction of length k.*

1. *For an input graph $\mathcal{G}$, the $(u, v)$-th entry of $O_k$ counts the number of groundings of $\phi$ in $\mathcal{G}$ where $U_1 = u$ and $V_P = v$. Therefore $n_\phi(\mathcal{G}) = \sum(O_k(\phi))$.*

2. *For an expected graph $\tilde{\mathcal{G}}_{\boldsymbol{z}}$, the $(u,v)$-th entry of $O_k$ counts the expected number of groundings of $\phi$ where $U_1 = u$ and $V_P = v$. Therefore $n_\phi(\tilde{\mathcal{G}}_{\boldsymbol{z}}) = \sum(O_k(\phi))$.*

*Proof.* We give the proof for clause 1, counting observed counts in an input graph $\mathcal{G}$. The argument for expected counts computed from an expected graph is exactly parallel.

Base case, $k = 1$. If $\phi = \{\ell(U,V)\}$, then the conjunction count is the number of pairs $(u,v)$ such that (i) both groundings $U = u$ and $V = v$ satisfy all unary literals, and (ii) $I_{\mathcal{G}}(\ell(U = u, V = V)) = 1$. All and only such pairs have the entry $\bar{\mathbf{A}}[u,v] = 1$ in the masked adjacency matrix associated with $\ell(U,V)$.

Case 1: $(U = V) \notin EQ$. Then the number of satisfying groundings is simply given by $\sum(\bar{\mathbf{A}})$.

Case 2: $(U = V) \in EQ$. Then the satisfying groundings are of the form $U = u, V = u$, so their count is given by the matrix trace of $\bar{\mathbf{A}}$, or equivalently $\sum(\bar{\mathbf{A}} \circ I)$. This establishes the base case.

Inductive Step: Assume the proposition holds for $k$ and consider the matrix $O_{k+1}$ computed by Equation (14). By the inductive hypothesis, the $(u,v)$-th entry of $O_k$ counts the number of instantiations of length $k$ between vertices $u$ and $v$ that satisfy $\phi'$. Now, the number of instantiations of length $k+1$ between $u$ and $w$ equals the number of instantiations of length $k$ from vertex $u$ to each vertex $v$ that has $\ell_{k+1}$ relation with $w$. The non-zero entries of column $w$ of masked matrix $\bar{\mathbf{A}}_{k+1}$ represent $v$s related by $\ell_{k+1}$ to $w$. So, $(u,w)$-th entry of $O_k\bar{\mathbf{A}}_{k+1}$ gives the number of instantiations between $u$ and $w$ satisfying the centered conjunction and all equality constraints except possibly $U_1 = V_{k+1}$. Therefore for the case where $U_1 = V_{k+1} \notin EQ$, the matrix $O_{k+1}$ satisfies the inductive hypothesis. For the case where $U_1 = V_{k+1} \in EQ$, we observe that the number of instantiations of length $k+1$ from node $u$ to $u$ equals the $(u,u)$ diagonal entry of $O_k\bar{\mathbf{A}}_{k+1}$ or equivalently, $(O_k\bar{\mathbf{A}}_{k+1}) \circ I$. Thus the total number of satisfying groundings is given by $\sum(O_{k+1}(\phi))$ in either case, which establishes the inductive hypothesis. $\qquad\square$

**Extensions**

- Our counting method is based on a **sorting algorithm**. A sorting algorithm is a procedure that takes a set of relationship literals

$$\ell_1(U_1, V_1), \ldots, \ell_P(U_P, V_P)$$

and determines if there exists a permutation $\pi$ such that the literals can be arranged to form a *chain conjunction*. Specifically, it seeks to satisfy the equality constraints given by

$$V_{\pi(i-1)} = U_{\pi(i)} \quad \text{for } i = 2, \ldots, P.$$

**Example**

Consider the following conjunction of relationship literals:

$$AdvisedBy(Student, Professor),$$
$$TaughtBy(Course, Professor),$$
$$Registered(Student, Course)$$

This set of relationships is not a chain conjunction because there is no permutation of the literals that satisfies the necessary chain equality constraints. However, we can transform this into a chain conjunction using the reverse relations:

$$AdvisedBy(Student, Professor),$$
$$Teaches(Professor, Course),$$
$$TakeCourse(Course, Student)$$

In this case, the literals can be rearranged to form a chain conjunction, satisfying the equality constraints. Here *Teaches* is the reverse of *TaughtBy* and *TakeCourse* is the reverse of *Registered*.

- Our algorithm and proof can be extended to the case of *nested conjunctions*, which have no crossing equalities. Say that two variable equalities $U_{k_1} = V_{k_2}$ and $U_{k_3} = V_{k_4}$ **cross** if $k_1 < k_3 < k_2$ and $k_2 < k_4$. A nested chain is composed of centered chains, so our matrix multiplication algorithm can be used to recursively compute instance counts.

### A.10 GNN-BASED GRAPH REALISM METRIC FOR EVALUATING GRAPH GENERATION

How to quantitatively assess generated graphs has been studied in recent papers O'Bray *et al.* (2022); Thompson *et al.* (2022); Shirzad *et al.* (2022). The general approach proceeds in two stages:

1. For a graph $\mathcal{G}$, extract a real-valued **descriptor** vector $\phi(\mathcal{G})$.

2. Measure the similarity $\mu(\mathcal{G}, \hat{\mathcal{G}})$ of an observed graph $\mathcal{G}$ and a generated graph $\hat{\mathcal{G}}$ by applying a distance/kernel on their real-valued descriptors vectors $\phi(\mathcal{G})$ and $\phi(\hat{\mathcal{G}})$.

The similarity of a set of observed graphs and a set of generated graphs can be quantified as the similarity of their descriptor sets using Maximum Mean Distance (MMD). The SOTA descriptor function utilizes a *reference embedding network* GNN $\mathcal{E}$. The embedder $\mathcal{E}$ is obtained from random weights and is therefore independent of any of the models under evaluation.

We adapt the SOTA GNN-based approach to the setting of learning from a single training graph $D$ as follows. We compare the training graph to generated expected graphs $\tilde{\mathcal{G}}_1, \ldots \tilde{\mathcal{G}}_m$. An expected graph $\tilde{\mathcal{G}}_i$ is generated by sampling a node embedding $z_i$ from the prior distribution $p(z)$, then applying the decoder model eq. (5) to $z_i$. Following Thompson *et al.* (2022), we apply an embedder $\mathcal{E}$ with random weights to the training graph resp. generated expected graphs to obtain embeddings $e$ resp. $\hat{e}_1, \ldots, \hat{e}_m$. The random GNN option does not require multiple training graphs. The message-passing mechanism of GNNs naturally extends to weighted graphs, so we can apply the GNN embedder to expected graphs directly. To quantify the similarity of the generated embeddings $\hat{e}_1, \ldots, \hat{e}_m$ to the training graph embedding $e$, we utilize the MMD metric with a linear kernel, which is recommended by Thompson *et al.* (2022). We used their code to compute the MMD metric results.

### A.11 LEARNING CURVES

We report a learning curve experiment to examine the effect of rule knowledge on data efficiency. The idea is to simulate the impact of a domain expert providing the model with a strong set of rules. After learning an informative set of rules on the entire training graph, we sample $20\%$ of node labels as test labels and reserve the other $80\%$ as training node labels. Then we sample $x\%$ of the training node labels for training the VGAE with and without rules. We report the predictive accuracy on the test labels, after training the VGAE with and without rules on $x = 25\%, 50\%, 75\%, 100\%$ of training labels. The models are tested on the remaining $100 - x\%$ of nodes. UW is too small to obtain a meaningful learning curve.

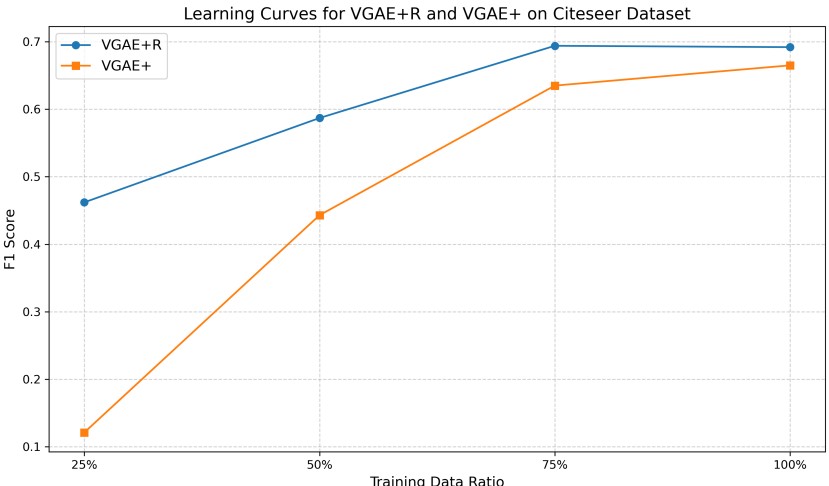

Figure 13: Learning curve for the CiteSeer dataset.

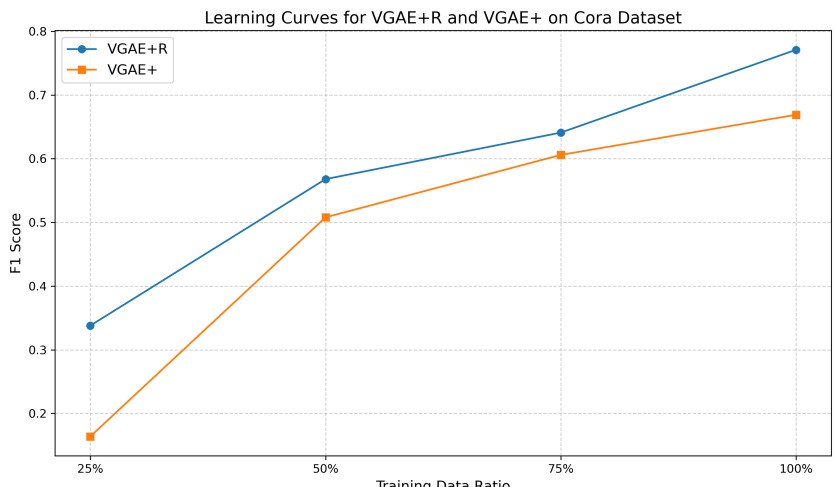

Figure 14: Learning curve for the Cora dataset.

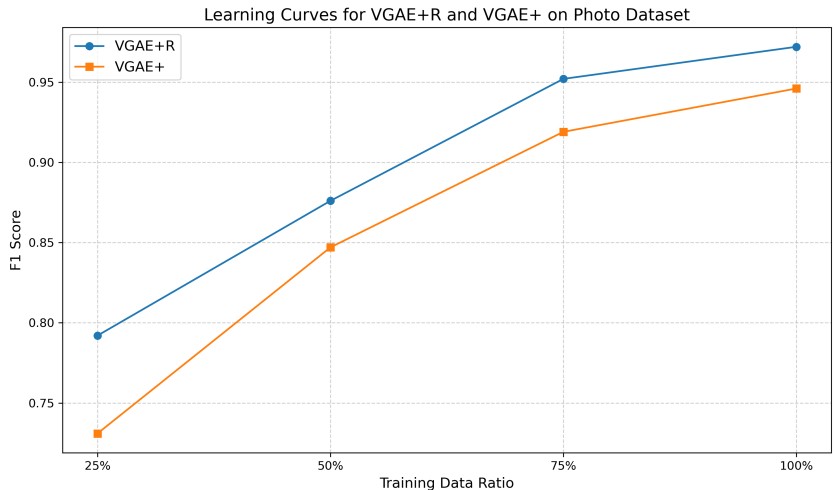

Figure 15: Learning curve for the Photo dataset.

### A.12 LOSS CURVES

In this section we show that rule matching has a big impact on training, by in effect initializing the GGM model in a part of weight space that encodes the rule knowledge.

#### A.12.1 LABEL LOSS CURVES

In this section, we compare the label loss component for each model across different datasets. As shown in Figures 19, 20, 21, 22, and 23, the label loss for VGAE+R is generally lower than that for VGAE+ across most datasets, suggesting that VGAE+R may yield better accuracy in node classification.

#### A.12.2 TOTAL LOSS CURVES

In this section, we present the total loss curves for both the VGAE+ and VGAE+R models across multiple datasets. While it is not possible to directly compare the loss values between the two models, the downward trend in loss during training for both models indicates successful convergence. Figures

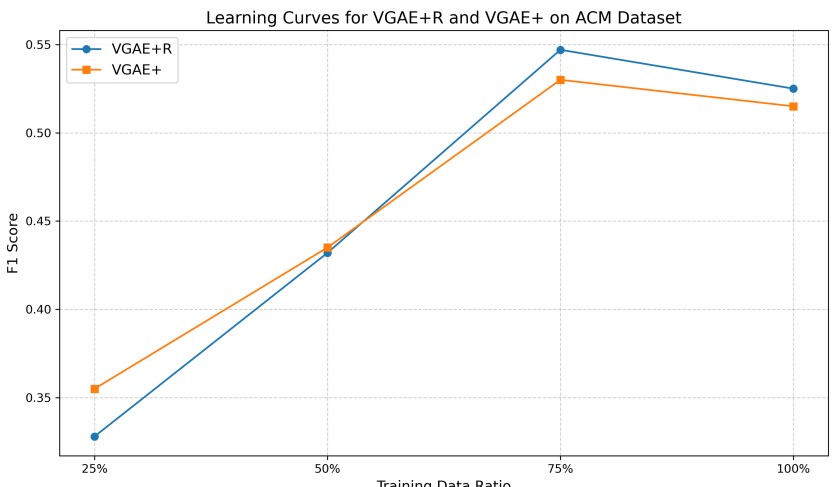

Figure 16: Learning curve for the ACM dataset.

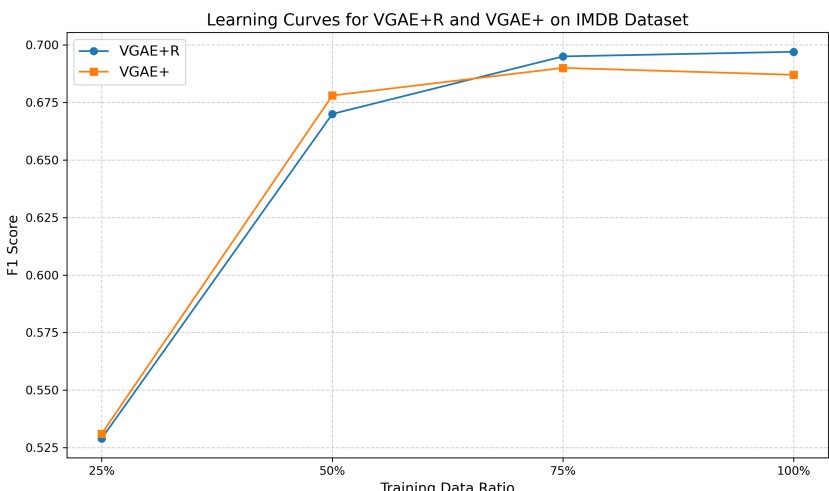

Figure 17: Learning curve for the IMDb dataset.

24, 25, 26, 27, and 28 illustrate the total loss for each dataset, confirming the models' progress throughout the training process.

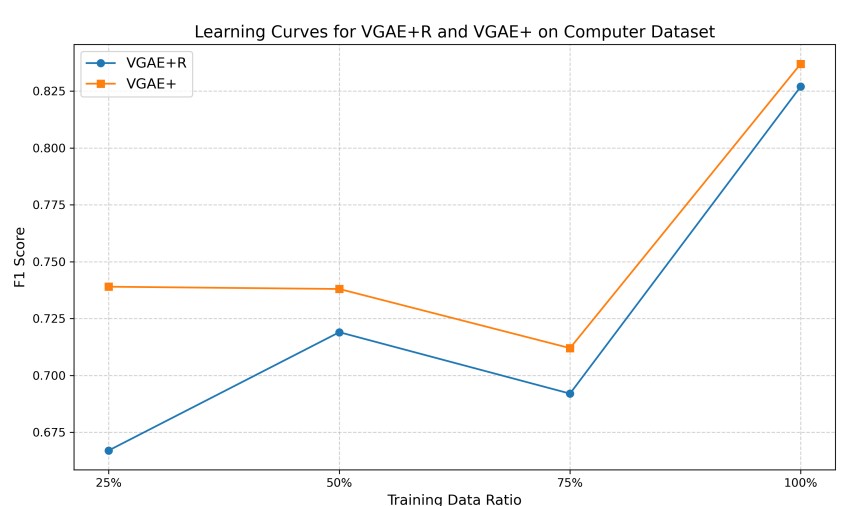

Figure 18: Learning curve for the Computers dataset.

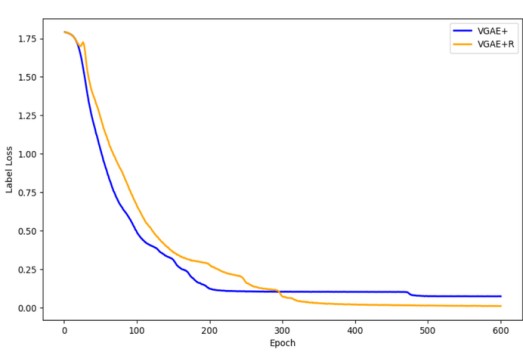

Figure 19: Label loss component for CiteSeer dataset during model training

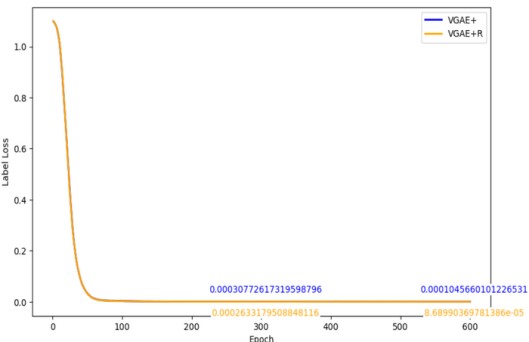

Figure 20: Label Loss component for ACM dataset during model training

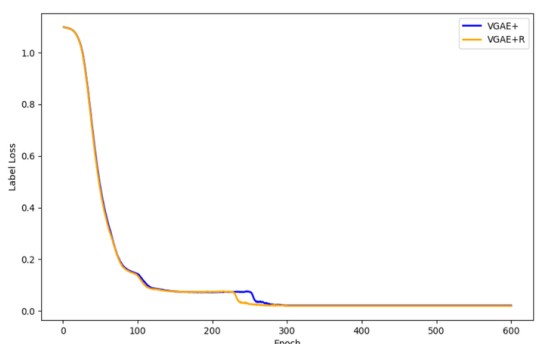

Figure 21: Label Loss component for IMDb dataset during model training

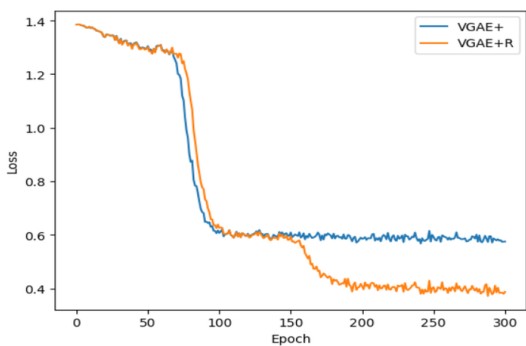

Figure 22: Label Loss component for UW dataset for person node type during model training

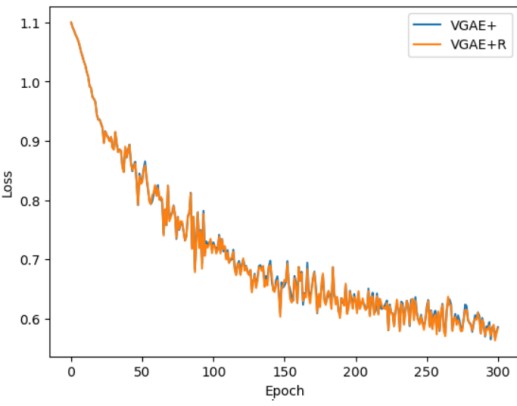

Figure 23: Label Loss component for UW dataset for course node type during model training

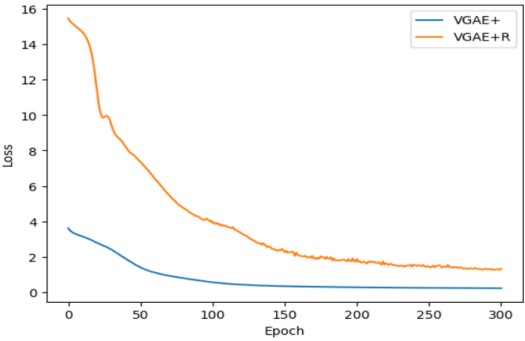

Figure 24: Total loss for Cora dataset during model training

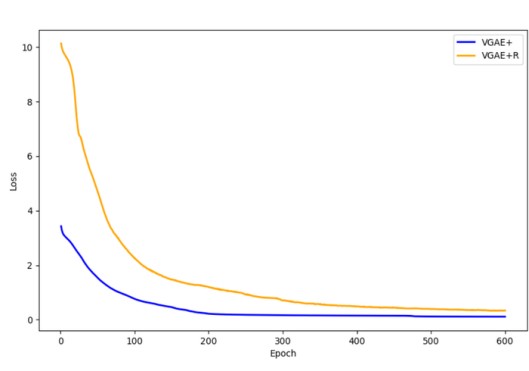

Figure 25: Total Loss for CiteSeer dataset during model training

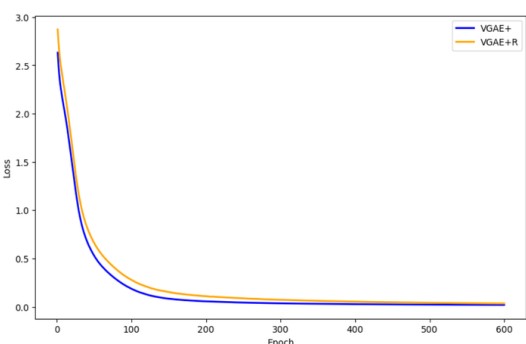

Figure 26: Total Loss for ACM dataset during model training

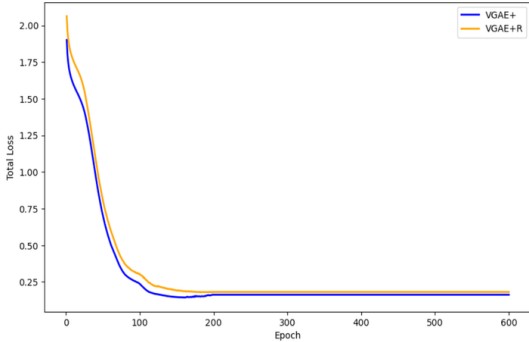

Figure 27: Total Loss for IMDb dataset during model training

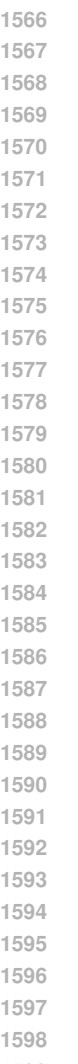

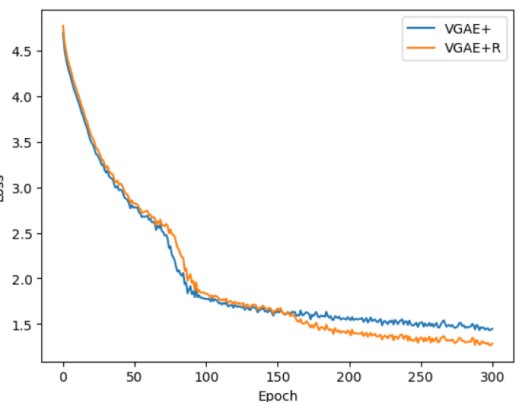

Figure 28: Total Loss for UW dataset during model training

Table 7: Graph quality comparison of VGAE+ and VGAE+R on six datasets using MMD metrics (degree, clustering, 4-orbit, spectral, diameter). Lower values indicate generated graphs are closer to observed ones; VGAE+R shows consistent improvements.

| Dataset | Metric | VGAE+ | VGAE+R |
|---|---|---|---|
| CiteSeer | Degree MMD | 0.752 | **0.604** |
| | Clustering MMD | **0.733** | 1.672 |
| | 4-Orbit MMD | 0.549 | **0.538** |
| | Spectral MMD | 0.506 | **0.431** |
| | Diameter MMD | 2.000 | **1.877** |
| Cora | Degree MMD | 0.768 | **0.707** |
| | Clustering MMD | 1.680 | **1.648** |
| | 4-Orbit MMD | 1.400 | **1.375** |
| | Spectral MMD | 0.514 | **0.461** |
| | Diameter MMD | 1.979 | **1.944** |
| IMDB | Degree MMD | 0.756 | **0.521** |
| | Clustering MMD | 1.950 | **1.870** |
| | 4-Orbit MMD | 0.599 | **0.523** |
| | Spectral MMD | 0.516 | **0.389** |
| | Diameter MMD | 1.894 | **1.760** |
| ACM | Degree MMD | 0.770 | **0.664** |
| | Clustering MMD | 1.930 | **1.880** |
| | 4-Orbit MMD | 0.756 | **0.692** |
| | Spectral MMD | 0.571 | **0.524** |
| | Diameter MMD | **1.944** | 1.955 |
| Photos | Degree MMD | 0.727 | **0.547** |
| | Clustering MMD | **1.760** | 1.810 |
| | 4-Orbit MMD | **1.090** | 1.260 |
| | Spectral MMD | 0.606 | **0.399** |
| | Diameter MMD | 1.941 | **1.305** |
| Computers | Degree MMD | 0.756 | **0.227** |
| | Clustering MMD | 1.952 | **1.867** |
| | 4-Orbit MMD | 0.788 | **0.767** |
| | Spectral MMD | 0.516 | **0.326** |
| | Diameter MMD | 1.894 | **0.706** |

### A.13 GRAPH REALISM EVALUATION

Our main metric of graph quality is the GNN-based graph realism MMD measure, which is the current state-of-the-art method for comparing generated graphs with observed graphs. This metric was introduced in prior work (Thompson *et al.*, 2022) and relies on a reference GNN $R$ to embed entire graphs. Importantly, $R$ is independent of the models under evaluation. In our experiments, we used a reference GNN with random weights, as recommended by Thompson *et al.* (2022), ensuring that no domain-specific information was encoded. A pre-trained reference GNN could also be used, but we did not adopt that setting in this paper.

For completeness, we also report statistic-based MMD measures (degree, clustering, 4-orbit, spectral, diameter), which have been widely used in earlier work on auto-regressive graph generation such as GraphRNN. These complementary metrics show consistent benefits from incorporating first-order rules.

Table 7 presents the performance of VGAE+ (baseline) and VGAE+R (rule-enhanced) across six benchmark datasets. Across both GNN-based and classical MMD measures, lower values indicate closer alignment between generated and observed graphs, and VGAE+R achieves lower scores in all but one setting (clustering on Citeseer), demonstrating the positive effect of rule-based enhancement.

Table 8: Count Distance between rule values in the test set and generated graphs. Lower values are better.

| Dataset | VGAE+ | VGAE+R |
|---------|-------|--------|
| Cora | 1403.530 | **18.020** |
| CiteSeer | 459.560 | **24.760** |
| IMDB | 25762.900 | **1082.700** |
| ACM | 474.360 | **118.900** |
| Photo | 6046.740 | **1212.110** |
| Computers | 29567.100 | **1006.970** |
| UW | 763257.950 | **744808.390** |

Table 9: Runtime and memory usage of VGAE+ and VGAE+R across benchmark datasets.

| Dataset | #Nodes | #Rules | VRAM (GB) | Runtime VGAE+ | Runtime VGAE+R | GPU |
|---------|--------|--------|-----------|---------------|----------------|-----|
| Cora | 2708 | 138 | 12.2 | 0m 22.9s | 0m 44.7s | GTX 1080 Ti |
| CiteSeer | 3327 | 193 | 11.7 | 1m 34.2s | 2m 18.9s | Titan X |
| IMDB | 12772 | 31 | 38.8 | 3m 36.7s | 4m 08.5s | A40 (48 GB) |
| ACM | 8993 | 62 | 41.2 | 2m 07.5s | 2m 34.4s | A40 (48 GB) |
| Photos | 7650 | 207 | 80.0 | 1m 36.1s | 2m 40.4s | A100 (80 GB) |
| Computers | 13752 | 58 | 79.5 | 4m 30.0s | 5m 24.2s | A100 (80 GB) |
| UW | 410 | 111 | 0.175 | 0m 15.0s | 0m 16.0s | GTX 1080 Ti |

### A.14 COUNT DISTANCE EVALUATION BASED ON PRIOR EMBEDDINGS

For the Count Distance metric, prior work on VAE-based graph generation has often evaluated fit to the training or test data using the ELBO as an indirect measure of generation quality (Simonovsky and Komodakis, 2018). In our setting, Count Distance similarly measures how well generated graphs reproduce rule instance counts observed in the data. One might expect a neural GGM to implicitly match motif counts as part of modeling the training distribution, but our results show that a standard VGAE does not. Adding our explicit semantic loss substantially reduces this discrepancy.

we also report results when sampling directly from the prior $p(z)$. As shown in Table 8, the semantic loss improves not only fit to training data but also pure generation under $p(z)$. Lower values indicate closer alignment between generated and observed rule counts, and VGAE+R consistently achieves much lower distances than VGAE across all datasets.

### A.15 SCALABILITY AND RUNTIME ANALYSIS

Our core algorithm relies on dense matrix multiplications over probabilistic adjacency matrices. As discussed in Section 5, this raises natural concerns about scalability. In this appendix, we provide additional details on the computational resources required and the observed runtime overhead of the semantic loss.

Table 9 summarizes, for each benchmark graph, the number of nodes, number of rules, peak GPU memory usage, runtime for 100 training epochs with VGAE+ and VGAE+R, and the GPU model used. Storage needs for generating graphs ranged from 1GB for the smallest dataset to 70GB for the largest, and end-to-end experiments (training + generation + evaluation) took between 4 and 5 hours for the biggest dataset.

Modern GPUs render dense matrix multiplications surprisingly efficient: thousands of cores execute multiply–add operations in parallel, while optimized libraries such as cuBLAS tile large matrices into on-chip shared-memory blocks. Tensor Cores and fused multiply–add instructions further accelerate throughput, often at reduced precision. As a result, the additional multiplications from our semantic loss add only a modest wall-clock overhead.

For example, runtime on the UW graph increases by just 1 second (15s to 16s). For mid-sized graphs such as ACM and IMDB, the slowdown is about 20% and 15%, respectively. Larger graphs such as Photos incur slowdowns of 1.95× and 1.67×, yet all runs complete in under six minutes. Even the

Table 10: Robustness of VGAE+R to noisy/incomplete rules on the Cora dataset.

| Condition | Degree | Clust. | 4-Orbit | Spectral | Diam. | CountDist (train) | CountDist (test) | Graph Realism | AUC (%) | F1 (%) |
|---|---|---|---|---|---|---|---|---|---|---|
| 50% Rules Deleted | 0.733 | 1.640 | 1.398 | 0.517 | **1.914** | 58991.850 | 35.190 | 1.745e20 | 94.5 | 85.0 |
| VGAE+R (full rules) | **0.707** | **1.648** | **1.375** | **0.461** | 1.944 | **26800.000** | **18.000** | 6.840e17 | **96.5** | **88.0** |
| Baseline VGAE+ | 0.768 | 1.680 | 1.400 | 0.514 | 1.979 | 51400.000 | 1403.000 | 1.400e19 | 86.0 | 69.0 |

13k-node Computers graph finishes in 5m 24s on an A100 GPU. On average, VGAE+R adds only a $1.4\times$ overhead while yielding substantial improvements in rule compliance and graph quality.

It is common practice to evaluate new objectives for graph generative models in settings that require a full dense adjacency matrix. Examples include GraphVAE (Simonovsky and Komodakis, 2018) and the more recent Digress diffusion model (Vignac *et al.*, 2023). These evaluations remain meaningful because many graph generation benchmarks consist of small to medium-sized graphs, such as molecules and proteins. A general lesson in the field is that all-at-once methods have fast training and generation for small to medium graphs, while scaling to very large graphs typically requires autoregressive approaches (Hamilton, 2020).

### A.16 ROBUSTNESS TO NOISY OR INCOMPLETE RULES

Almost all rules we use are non-deterministic, meaning they allow for exceptions (e.g., "if $X$ works in city $Y$, then $X$ lives in city $Y$," which may not always hold). This flexibility is part of the power of first-order rules, which can hold in a graph to a degree rather than absolutely (Domingos and Richardson, 2007). If noise refers to uninformative rule bodies, we prune such rules during preprocessing, which increases robustness. If the rule set is incomplete—missing important rules—the system degrades gracefully toward the standard GGM likelihood. In the extreme case of an empty rule set, our objective reduces exactly to the standard VGAE loss. Since our implementation derives rules from a first-order Bayesian network that represents the full joint domain distribution (Schulte and Qian, 2015), the resulting rule sets are likely close to complete.

Table 10 reports results on the Cora dataset under three settings: (1) deleting the top 50% of rules, (2) using the full VGAE+R rule set, and (3) the baseline VGAE+. Metrics include generation quality (MMD measures), count distance on train/test graphs, graph realism ($\times 10^n$), and classification accuracy (AUC, F1). Results confirm that pruning rules degrades performance, but VGAE+R remains robust compared to the baseline VGAE+.

