# OpenReview forum: "Enhancing Graph Generation With First-Order Logic Rules"
_ICLR.cc/2026/Conference — ICLR 2026 Conference Withdrawn Submission_

### Official Review · Reviewer_UNFG · 2025-10-27

**Soundness:** 2
**Presentation:** 3
**Contribution:** 2
**Rating:** 4
**Confidence:** 3

**Summary:**

While existing graph generation methods can achieve realistic results, the generated graphs lack domain-specific patterns. This paper proposes a method that uses first-order logic to enhance VGAE, making the domain-specific patterns in VGAE-generated graphs more closely match those in the original input graphs. The paper introduces a matrix multiplication-based approach to compute the number of domain-specific patterns in generated graphs, enabling their incorporation into the loss function. The authors validate their method on multiple datasets, demonstrating that it effectively improves the quality of generated graphs compared to the original VGAE.

**Strengths:**

1. The paper proposes that generated graphs and original input graphs should have a similar number of rules, which is an intuitive and easily understandable idea.
2. The paper proposes an automated feature extraction method, which is valuable.
3. To efficiently compute the number of instances of first-order logic rules, the paper develops a novel matrix multiplication-based algorithm. This algorithm can be applied to standard graph generation hybrid models, and for conjunctive formulas that satisfy mild syntactic constraints, the number of instances can be computed through a series of adjacency matrix multiplication operations.
4. The paper thoroughly discusses the limitations/shortcomings of its own method.

**Weaknesses:**

1. The paper only uses the original VGAE as baseline, while the current mainstream methods for graph generation are diffusion methods and flow matching methods. Although the authors' intention may be to enhance VGAE, I believe it is necessary to compare with the current mainstream diffusion and flow matching methods. In DiGress[1] and CatFlow[2], it is mentioned that on molecular graphs, the validity of results generated by flow matching and diffusion methods is superior to that of ordinary VGAE (if I understand correctly, validity should correspond to the domain-specific patterns in this paper). If the proposed method cannot outperform flow matching and diffusion methods in this aspect, I believe this paper's method lacks practical value, or the authors should demonstrate the advantages of VGAE over flow matching and diffusion methods.

2. In the GRAPH REALISM experiments, the paper directly uses MMD as the metric. Such enormous values like 4.03e18 and 6.84e17 are difficult to understand, making it impossible to know what level the original VGAE's GRAPH REALISM is at, and it's also hard to understand how low the MMD of VGAE+R actually is. I suggest using some completely random method as a baseline, using that method's MMD value as a reference point, and normalizing the results of other methods accordingly to facilitate understanding of the results.

[1] Vignac, Clement, et al. "Digress: Discrete denoising diffusion for graph generation." arXiv preprint arXiv:2209.14734 (2022).
[2] Eijkelboom, Floor, et al. "Variational flow matching for graph generation." Advances in Neural Information Processing Systems 37 (2024): 11735-11764.

**Questions:**

1. The paper's motivation is that while existing graph generation methods can achieve realistic results, the generated results lack domain-specific patterns and are not semantically valid enough. I don't quite understand what the difference is between realistic and semantically valid?
2. In the matrix calculation examples in the paper, all matrix elements are binary. In practice, the elements of this matrix should be continuous values. Can you provide an example with continuous values? To be honest, your method is not explained clearly enough, and a more practical example would help better understand your approach.

---

> ### Author Response · Authors · 2025-11-30
> **Response to reviewer UNFG**
>
> Thank you for the time you spent reading our submission and writing your comments. We answer your questions starting with the most critical questions first.
>
> **The paper only uses the original VGAE as baseline, while the current mainstream methods for graph generation are diffusion methods and flow matching methods. Although the authors' intention may be to enhance VGAE, I believe it is necessary to compare with the current mainstream diffusion and flow matching methods.**
>
> We discuss the reasons for our choice of VGAE under Related Work (line 125) and note that extensions to other models for graph generation are a fruitful avenue for future reserach. The main issue is an impedance mismatch: our goal is to leverage rule learning for graph generation, but current rule learners *are limited to single graph inputs only*. So we need a graph generative model that applies to single graph, and the only candidate is the VGAE (see our Related Work section). Extending rule learners to multiple graphs is a difficult open problem that we did not want to solve in this paper. In fact, one of us served on the Neurips program committee this year and reviewed a submission that aimed to solve rule learning for multiple graphs.
>
>  **I don't quite understand what the difference is between realistic and semantically valid?** Realistic implies semantically valid but not the other way around. A semantic valid model of traffic lights restricts them to green, yellow, red but satisfying that constraint is not enough to generate realistic traffic light behavior.
>
> **I suggest using some completely random method as a baseline, using that method's MMD value as a reference point, and normalizing the results of other methods accordingly to facilitate understanding of the results.** Well we do report percentage improvements over the baseline. The standard method for the MMD metric (see our references) is to split your set of training graphs randomly 50-50 into training/test split and then compute the MMD between the training and test graphs as an ideal lower bound. We cannot do that because (again), we are learning from a single graph only.
>
> **In the matrix calculation examples in the paper, all matrix elements are binary. In practice, the elements of this matrix should be continuous values. Can you provide an example with continuous values? To be honest, your method is not explained clearly enough, and a more practical example would help better understand your approach.**
>
> Adjacency matrices are inherently binary, so readers familiar with graph representations will find binary matrices most natural. That said, the expected graph *is* continuous and Figure 12 in the appendix illustrates the matrix multiplication method with continous matrices. Perhaps you missed that figure since it is in the appendix. The appendix section would probably have clarified our method for you because the main text is necessarily compressed at a high level due to the page limit.

---

### Official Review · Reviewer_abnm · 2025-10-28

**Soundness:** 2
**Presentation:** 2
**Contribution:** 2
**Rating:** 4
**Confidence:** 3

**Summary:**

The authors add a first-order (FO) “semantic loss” that moment-matches rule instance counts while training a graph generator; compute rule counts via matrix multiplications on (soft) adjacency/features.

**Strengths:**

1. The proposed model establishes a clear and principled connection between SRL/MLN “moment matching” and modern GGM training. In the special case, the FO loss simplifies to propositional semantic loss.

2. Develop a practical counting routine for chain-conjunctive formulas that can be differentiated on probabilistic graphs.

**Weaknesses:**

1. The primary training objective is classic MLN/maximum-entropy moment matching with a neural likelihood term. The novelty lies mainly in an efficient instantiation for centered chain conjunctions using matrix products. It’s uncertain how broadly this extends beyond chains without combinatorial blow-ups or ad-hoc handling. The paper formalizes the chain case (Prop. 3) but lacks comparable rigor for general FO motifs.
2. Authors (i) learn rules from the training graph (Factorbase), (ii) regularize the generator to match those rule counts, and (iii) evaluate using GNN-MMD with a random reference network. This approach carries the risk of circularity: the model is intentionally nudged to replicate the statistics that the rules capture. A random-weight embedder might inadvertently favor these count-aligned patterns, leading to inflated MMD improvements. The paper only considers random-weight R, neglecting stronger alternatives (e.g., pretrained task-agnostic embedders) that might be less susceptible to a specific set of motifs. It is crucial to provide a justification for why random-GNN MMD is the appropriate arbiter in this context and to include complementary evaluations beyond MMD.
3. The authors introduce self-loops and symmetrize edges to facilitate message passing. However, it’s crucial to determine whether the counts are matched on the preprocessed graph or the original graph. This distinction can significantly impact motif counts, particularly for triangles and cycles, thereby altering the effective constraint. Therefore, it’s essential to standardize and document the counting domain.

**Questions:**

1. Are rule counts matched on the original graph or on the preprocessed (self-looped, undirected) graph used for message passing?
2. How robust are results to noisy/misaligned rules? (E.g., random or adversarial rules with high BIC score due to spurious correlations.)

---

> ### Author Response · Authors · 2025-11-30
> **Response to reviewer abnm**
>
> Thank you for the time you spent reading our submission and writing your comments. We answer your questions starting with the most critical questions first.
>
> **Are rule counts matched on the original graph or on the preprocessed (self-looped, undirected) graph used for message passing?** They are matched on the original graph. The preprocessed graph is used only for message passing. Enhancing the original training graph for message passing is common in graph neural networks (see e.g., graphnomers, graph expanders).
>
> **A random-weight embedder might inadvertently favor these count-aligned patterns, leading to inflated MMD improvements. The paper only considers random-weight R, neglecting stronger alternatives (e.g., pretrained task-agnostic embedders) that might be less susceptible to a specific set of motifs. It is crucial to provide a justification for why random-GNN MMD is the appropriate arbiter in this context and to include complementary evaluations beyond MMD.** Well random-weight GNNs for computing reference embeddings *are* recommended in the references we give. Random weights are definitely task-agnostic (in fact, they are dataset agnostic). Pre-training GNNs is more likely to lead to reference embeddings that incorporate bias towards motifs in our opinion. That said, we accept that it may be possible to mitigate such bias through "task-agnostic" training. The other problem is that, as with the over reviewers, pre-training requires a *set* of training graphs and we are learning rules from a single graph since that is what rule learners support.
>
> **How robust are results to noisy/misaligned rules? (E.g., random or adversarial rules with high BIC score due to spurious correlations.)** That is a good question and we mention the reliance on good rules under Limitations. We did do some experiments (not reported in the paper). Briefly not pruning rules leads to slightly higher accuracy on downstream tasks but significantly higher computational cost.

---

### Official Review · Reviewer_Cj78 · 2025-11-01

**Soundness:** 2
**Presentation:** 3
**Contribution:** 4
**Rating:** 4
**Confidence:** 3

**Summary:**

The paper proposes a semantic loss to be combined with a variational graph auto-encoder (VGAE) in order to inject domain information expressed in the form of first-order logical rules. The key aspects of the loss are that (1) is efficient to compute in practice, and (2) maximizes the training graph's likelihood while minimizing a divergence between observed and expected instance counts. This loss is claimed to improve the quality of the graphs generated by the method, when measured using graph motif counts or by comparing graph embeddings.

**Strengths:**

I believe the paper does a good job at explaining the proposed method and why it should bring improvements while being efficient. Although I think some parts are missing some important details and notation is unclear (see weaknesses), I found the paper to be sufficiently clear and easy to understand.

I have found the experimental setting sufficiently clear. However, my expertise does not enable me to verify whether the large numbers reported in Table 2 for count distance and graph realism make sense (see my discussion in weaknesses).

**Weaknesses:**

Although I have appreciated the writing, I think the notation is often unclear.
- In L248 $p_{\eta}(A\mid z)$ is said to be a function from $\mathbb{R}^d\times\mathbb{R}^d$ to $[0,1]$. To my understanding, $p_{\eta}$ defines the likelihood _of a single link_ using the node embeddings. However, this is contrast with the notation where the distribution is conditioned on some latent vector $z$. Is $z$ denoting the embeddings of _all the nodes_? For similar reasons, the notation used for feature and label decoders is not clear.
- Are the summations in L19 and L20 of Algorithm 1 over all the entries of the matrix? I think adding comments in the algorithm would help.
- L319. It is not clear what symbols A, X, L with tildes refer to.

I believe the authors could have detailed how their work and the literature about probabilistic databases are related. See e.g. [A] and [B], which are not cited. Although the authors mention the work [C], I was not able to understand how Eq. 1 is directly related to model counting. That is, model counting requires us to sum the weights taken over _all possible worlds_ that are consistent with the logical formula. Eq. 1 only considers conjunctive formulae and the sum is defined over all possible assignments _of only the variables that appear in the logical formula_, which usually consists of only a subset of the relation symbols. Could you please clarify the relationship between the probabilistic instance count (Eq. 1) and the works [A], [B] and [C]?

[A] Nilesh N. Dalvi, Dan Suciu. Efficient Query Evaluation on Probabilistic Databases. 2004.
[B] Nilesh N. Dalvi, Dan Suciu. The dichotomy of probabilistic inference for unions of conjunctive queries. 2012.
[C] Ondrej Kuzelka. Counting and sampling models in first-order logic. In International Joint Conference on Artificial Intelligence. 2023.

In L258-259 the authors mention that "FO moment matching can be implemented by performing (probabilistic) instance counting in a single graph". I found this sentence very confusing, as there is not a single graph, but rather a distribution over graphs. The authors say only later in L318-319 and Appendix A.9.2. that the algorithm is run with the probabilistic adjacency matrices, i.e., with entries between 0 and 1.

In Table 2 the authors show average percentage improvements. Standard deviations are only reported for raw count distance values. I am confident with the significance of the shown results, as the standard deviations are very high (and sometimes close to the means as for the IMBDb dataset). Although the authors mentions some works about the graph realism metric (L364), I could not find previous results having the same order of magnitudes, which arises doubts about how these metrics have been calculated and used.

The lack of clarity in certain parts about the methodology and about some of the reported numbers is what makes me lean towards a much lower score. I am happy to reconsider these aspects during the rebuttal.

**Questions:**

See my questions and points made in the weaknesses section.

---

> ### Author Response · Authors · 2025-11-29
> **Response to reviewer Cj78**
>
> Thank you for the time you spent reading our submission and writing your comments. We answer your questions starting with the most critical questions first.
>
>
>
> 1. **In L258-259 the authors mention that "FO moment matching can be implemented by performing (probabilistic) instance counting in a single graph". I found this sentence very confusing, as there is not a single graph, but rather a distribution over graphs. The authors say only later in L318-319 and Appendix A.9.2. that the algorithm is run with the probabilistic adjacency matrices, i.e., with entries between 0 and 1.**
>
> Too bad, we meant for this sentence to be a simplifying paraphrase of the formal statement in Proposition 2. There is indeed a distribution over graphs; moment matching requires computing the *expected value* of rule instance counts wrt this distribution. The "single graph" we refer to is the *expected graph* with respect to this distribution. Prop. 2 says that it suffices to perform probabilistic instance counting in the expected graph. See Prop. 2 for definition and Figures 2 and 3 for illustration.
>
> We agree that it is not obvious that computing an expectation wrt a distribution over all graphs can be done wrt a single graph-this is why Proposition 2 is highly nontrivial and we listed it as a main contribution! The result belongs to a general family of results along the lines of "expected value of a function = function applied to expected input" (think linear functions or Jensen's inequality).
>
> 2.  **I was not able to understand how Eq. 1 is directly related to model counting**. Well in propositional model counting, we count the number of groundings that satisfy a given formula. In weighted model counting, each grounding is assigned a weight, and we compute the weighted sum of groundings that satisfy the formula. In equation 1, the innner product loop computes the weight of conjunctive formula $\phi$, and the outer loop computes the weighted sum. Hence what we call "probabilistic instance count" is the weighted model count of $\phi$. We agree that connecting with the literature on WMC and probabilistic databases is a good idea (assuming we have the space). That is why we noted this as a direction for future work. For our paper, the connection with generative graph models is the most relevant.
>
>  3. **In L248 $p_{\eta}(\mathbf{A}|\mathbf{z})$  is said to be a function from  to $R^{d} \times R^{d} $ to $[0,1]$.  To my understanding,  defines the likelihood of a single link using the node embeddings. However, this is contrast with the notation where the distribution is conditioned on some latent vector . Is  denoting the embeddings of all the nodes? For similar reasons, the notation used for feature and label decoders is not clear.**
>
> Right, $p_{\eta}(\mathbf{A}|\mathbf{z})$ denotes a distribution over adjacency matrices that is induced by the link decoder if we treat all links as (conditionally) independent. we agree this is compressed notation. Yes $\mathbf{z}$ refers to a vector of all node embeddings.

---

### Note · Authors · 2025-12-16

I have read and agree with the venue's withdrawal policy on behalf of myself and my co-authors.